# Ndfip1 restricts mTORC1 signalling and glycolysis in regulatory T cells to prevent autoinflammatory disease

Awo Akosua Kesewa Layman[1,2,*], Guoping Deng[3,*], Claire E. O'Leary[3,*], Samuel Tadros[3], Rajan M. Thomas[4], Joseph M. Dybas[4], Emily K. Moser[3], Andrew D. Wells[4], Nicolai M. Doliba[5] & Paula M. Oliver[3,4]

Foxp3[+] T regulatory ($T_{reg}$) cells suppress immune cell activation and establish normal immune homeostasis. How $T_{reg}$ cells maintain their identity is not completely understood. Here we show that Ndfip1, a coactivator of Nedd4-family E3 ubiquitin ligases, is required for $T_{reg}$ cell stability and function. *Ndfip1* deletion in $T_{reg}$ cells results in autoinflammatory disease. Ndfip1-deficient $T_{reg}$ cells are highly proliferative and are more likely to lose Foxp3 expression to become IL-4-producing $T_H2$ effector cells. Proteomic analyses indicate altered metabolic signature of Ndfip1-deficient $T_{reg}$ cells and metabolic profiling reveals elevated glycolysis and increased mTORC1 signalling. Ndfip1 restricts $T_{reg}$ cell metabolism and IL-4 production via distinct mechanisms, as IL-4 deficiency does not prevent hyperproliferation or elevated mTORC1 signalling in Ndfip1-deficient $T_{reg}$ cells. Thus, Ndfip1 preserves $T_{reg}$ lineage stability and immune homeostasis by preventing the expansion of highly proliferative and metabolically active $T_{reg}$ cells and by preventing pathological secretion of IL-4 from $T_{reg}$ cells.

[1] Medical Scientist Training Program, Perelman School of Medicine at the University of Pennsylvania, Philadelphia, Pennsylvania 19104, USA. [2] Biomedical Graduate Studies, Perelman School of Medicine at the University of Pennsylvania, Philadelphia, Pennsylvania 19104, USA. [3] Cell Pathology Division, The Children's Hospital of Philadelphia, Philadelphia, Pennsylvania 19104, USA. [4] Department of Pathology and Laboratory Medicine, The Children's Hospital of Philadelphia, Philadelphia, Pennsylvania 19104, USA. [5] Institute for Diabetes, Obesity, and Metabolism, University of Pennsylvania School of Medicine, Philadelphia, Pennsylvania 19104, USA. * These authors contributed equally to this work. Correspondence and requests for materials should be addressed to P.M.O. (email: paulao@mail.med.upenn.edu).

Foxp3[+] regulatory T ($T_{reg}$) cells suppress spontaneous immune cell activation and limit effector cell function, thereby preventing autoimmune and inflammatory disorders[1,2]. While generally stable, $T_{reg}$ lineage cells can have a high degree of instability in inflammatory settings. $T_{reg}$ cell instability is characterized by the loss of suppressive function, loss of Foxp3 protein or gain of pro-inflammatory cytokine production[3,4]. Defining pathways that help to establish and maintain Foxp3 expression, promote $T_{reg}$ cell suppressive function, prevent $T_{reg}$ cell production of pro-inflammatory cytokine and/or maintain $T_{reg}$ cell numbers will aid in the development of new $T_{reg}$ cell-based therapeutic applications.

Appropriate regulation of cellular energetics and metabolism is important for $T_{reg}$ cell function and lineage stability[5]. Unlike effector T cells, which rely heavily on glycolysis, $T_{reg}$ cells rely on fatty acid oxidation for their energy needs[6]. Mechanistic target of rapamycin (mTOR) is a serine–threonine kinase that forms part of the mTORC1 and mTORC2 protein complexes and is a critical regulator of cellular metabolic processes. Both complexes can limit glycolysis in $T_{reg}$ cells thereby promoting lineage stability and suppressive functions[7,8]. Although metabolic state is clearly important for $T_{reg}$ maintenance and function, many factors that impact $T_{reg}$ cell metabolism remain unknown.

Ubiquitylation is a fundamental post-translational modification affecting many aspects of T-cell differentiation and function[9,10]. Neural precursor cell expressed, developmentally downregulated 4 (Nedd4) family interacting protein 1 (Ndfip1) is a transmembrane protein that binds and activates Nedd4 family E3 ubiquitin ligases[11]. The highly conserved catalytic E3 ligases perform two functions in protein ubiquitylation: binding to the specific ubiquitylation target and catalysing the final transfer of ubiquitin. Ndfip1 activation of the Nedd4 E3 ligase Itch results in ubiquitylation and degradation of the transcription factor JunB, thereby limiting interleukin (IL)-4 cytokine production from T helper type 2 ($T_H2$) cells and $T_H2$-mediated inflammatory disease[12,13].

Ndfip1-deficient mice have decreased $T_{reg}$ cell numbers in the small bowel, a site of peripheral $T_{reg}$ generation, likely due to increased IL-4 signalling, which is inhibitory to $T_{reg}$ differentiation[14]. However, whether Ndfip1 also modulates $T_{reg}$ function after cells have committed to the $T_{reg}$ cell lineage has not been explored. Given that $T_{reg}$-specific deletion of Itch results in a $T_H2$-biased autoinflammatory disease[15], it seems plausible that Ndfip1 might be required to support Itch function in $T_{reg}$ cells.

Here we show that Ndfip1 expression in $T_{reg}$ cells prevents spontaneous inflammation at several sites, such as the lungs and skin. Ndfip1 limits both the accumulation and proliferation of CD44[+] effector $T_{reg}$ cells and prevents $T_{reg}$ cell production of IL-4. Consistent with increased proliferation and exposure to IL-4, $T_{reg}$ cells lacking Ndfip1 show increased conserved non-coding DNA sequence 2 (CNS2) methylation and are prone to losing Foxp3 expression in vivo. Increased T-cell proliferation is associated with increased mTORC1 signalling and high glycolytic activity, metabolic programmes that can fuel effector function in $T_{reg}$ cells and contribute to $T_{reg}$ lineage instability. Thus Ndfip1 maintains lineage identity in $T_{reg}$ cells and prevents these cells from aberrant acquisition of effector T-cell function. Ndfip1 is therefore a critical molecular sentinel that prevents autoinflammatory disease.

## Results

**$T_{reg}$-specific loss of Ndfip1 results in inflammation.** $Ndfip1^{-/-}$ mice develop a severe autoinflammatory disease by 6 weeks of age, resulting in death[12,16]. While thymic $T_{reg}$ output in Ndfip1-deficient mice is not altered[14], $T_{reg}$ cell numbers are reduced at the sites of peripheral $T_{reg}$ induction[14]. Therefore, to test the role of Ndfip1 within committed $T_{reg}$ cells, we generated mice in which $Ndfip1$ is conditionally deleted in $T_{reg}$ cells using the $Foxp3$-Cre-YFP reporter mice[17]. We observed that $Ndfip1$ mRNA is induced upon stimulation of control Ndfip1[+/+] $Foxp3$-Cre-sorted YFP[+] $T_{reg}$ cells and that $Ndfip1$ message is effectively ablated in $T_{reg}$ cells from $Ndfip1^{fl/fl}$ $Foxp3$-Cre male mice, which lack Ndfip1 in all $T_{reg}$ cells (Fig. 1a).

By 9–16 weeks of age, $Ndfip1^{fl/fl}$ $Foxp3$-Cre male mice developed pronounced splenomegaly, lymphadenopathy and progressive dermatitis (Fig. 1b). Histologically, the skin, oesophagus and lung showed marked immune infiltration inflammation (Fig. 1b). Analysis of spleen weight-to-body weight ratios revealed that male $Ndfip1^{fl/fl}$ $Foxp3$-Cre mice showed evidence of lymphoid expansion (Fig. 1c). To determine which immune responses were contributing to the observed inflammation, we examined serum immunoglobulin (Ig) levels. We found elevated levels of IgE and IgG1, indicative of type 2 inflammation, as well as elevated IgM (Fig. 1d). To determine the contribution of CD4 T cells to this pathology, we examined the spleen and lung of the mice and found increased activated phenotype (CD44[+]) CD4 T cells (Fig. 1e). CD4 T cells present in the lung of male $Ndfip1^{fl/fl}$ $Foxp3$-Cre animals were more likely to express the effector cytokines interferon-γ (IFNγ), IL-4 and IL-17A upon ex vivo stimulation (Fig. 1f). These data suggested that $Ndfip1$ expression in $T_{reg}$ cells is required for suppression of tissue inflammation and pathology.

**$Ndfip1^{fl/fl}$ $Foxp3$-Cre mice have more CD44[+] e$T_{reg}$ cells.** The observed pathology could result from a loss of $T_{reg}$ cell numbers (as occurs with mice lacking Foxp3[18]) or $T_{reg}$ cell function (as is seen in $T_{reg}$ cell-specific CTLA4 deficiency[19]). We therefore examined $T_{reg}$ percentages and numbers in the spleens and lung of $Ndfip1^{fl/fl}$ $Foxp3$-Cre and controls. Surprisingly, $T_{reg}$ cell numbers were increased in $Ndfip1^{fl/fl}$ $Foxp3$-Cre animals (Fig. 2a–c). Further, analysis of the $T_{reg}$ cell effector proteins— inducible T-cell costimulator (ICOS), programmed cell death-1 (PD-1) and glucocorticoid-induced TNFR-related protein (GITR)—revealed increased expression on $T_{reg}$ cells from $Ndfip1^{fl/fl}$ $Foxp3$-Cre male mice compared to Cre[+] controls (Fig. 2d). We did not see marked changes in CD25 levels. Taken together, these data suggest that the immunopathology observed in $Ndfip1^{fl/fl}$ $Foxp3$-Cre male mice is not due to decreased $T_{reg}$ numbers or loss of effector proteins known to support $T_{reg}$ function.

$T_{reg}$ cells can be quiescent or activated; these two subsets can be distinguished by the expression of CD44 and CD62L (ref. 20). Analysis of the lung of male $Ndfip1^{fl/fl}$ $Foxp3$-Cre mice revealed an increase in frequency (Fig. 2e,f) and numbers (Fig. 2g) of Foxp3[+] cells with an activated or 'effector' (e$T_{reg}$) phenotype (CD62L[lo]CD44[+]). In contrast, $Ndfip1^{fl/fl}$ $Foxp3$-Cre and control mice contained equivalent numbers of cells with a quiescent or 'central' (c$T_{reg}$) phenotype (CD62L[hi]CD44[−]) (Fig. 2g). ICOS, PD-1 and GITR are more highly expressed on e$T_{reg}$ cells. In contrast, CD25 is particularly high on c$T_{reg}$ cells[20]. Thus the observed increase in surface expression of ICOS, PD-1 and GITR observed on total $T_{reg}$ cells from $Ndfip1^{fl/fl}$ $Foxp3$-Cre animals could be due to an increased proportion of e$T_{reg}$ cells. To assess this, we examined the surface expression of ICOS, GITR, CD25 and PD-1 on the e$T_{reg}$ and c$T_{reg}$ populations. c$T_{reg}$ cells, as expected, had low levels of ICOS, GITR and PD-1, and loss of Ndfip1 did not alter this (Fig. 2h). However, c$T_{reg}$ cells from $Ndfip1^{fl/fl}$ $Foxp3$-Cre mice had decreased levels of CD25. Strikingly, in addition to their increased frequency, e$T_{reg}$ cells from $Ndfip1^{fl/fl}$ $Foxp3$-Cre mice showed higher levels of ICOS, GITR, CD25 and PD-1 compared to wild-type (WT) counterparts (Fig. 2h).

$cT_{reg}$ cells are thymically derived and undergo peripheral conversion to $eT_{reg}$ cells under the instruction of T-cell receptor (TCR) stimulation and ICOS stimulation[20]. Therefore, observing increased frequency and number of $eT_{reg}$ cells could indicate either increased conversion of $cT_{reg}$ cells or expansion of $eT_{reg}$ cells. An increased $cT_{reg}$ conversion rate would be expected to decrease the numbers of $cT_{reg}$ cells, unless compensated for by thymic output. As there was no decrease in the peripheral numbers of $cT_{reg}$ cells in $Ndfip1^{fl/fl}$ $Foxp3$-Cre male mice, we next examined thymic output utilizing mixed bone marrow chimera animals in which WT ($Ndfip1^{+/+}$ $Foxp3$-Cre)

and $Ndfip1^{fl/fl}$ $Foxp3$-Cre $T_{reg}$ cells develop in the same environment to control for effects of inflammation. In control and mixed chimeras, total $T_{reg}$ thymic output was unchanged (Supplementary Fig. 1a). Furthermore, within the mixed chimeras there was no significant difference in the percentge of thymic $T_{reg}$ cells derived from Ndfip1-sufficient or -deficient cells relative to the observed reconstitution ratio of all thymic $CD4^+$ T cells (Supplementary Fig. 1b). This suggests that the increase in $eT_{reg}$, but not in $cT_{reg}$, cell number in the periphery is not due to an altered rate of $cT_{reg}$ cell conversion to $eT_{reg}$ cells.

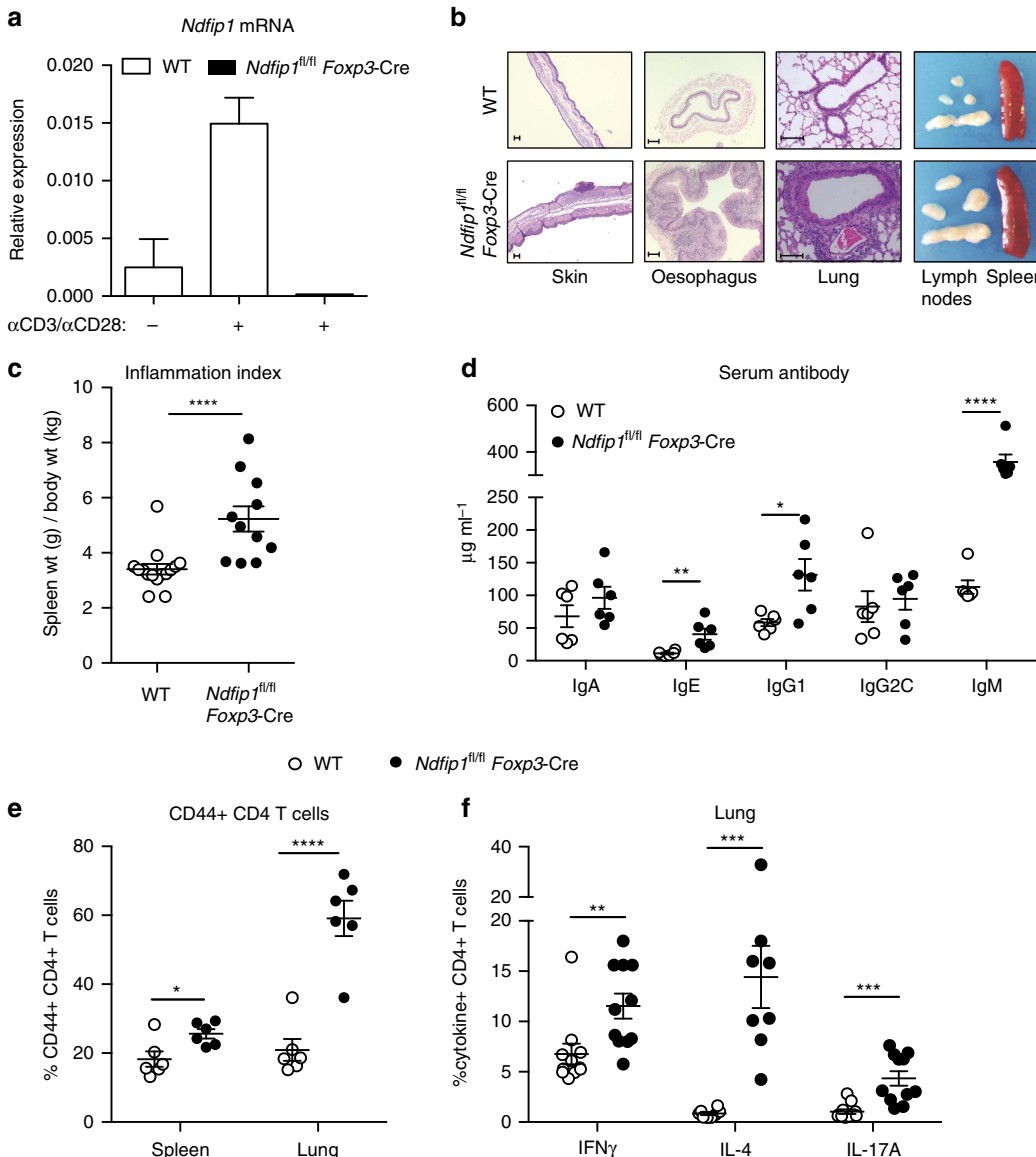

**Figure 1 | Mice lacking Ndfip1 in their $T_{reg}$ cells develop inflammatory disease.** (**a**) *Ndfip1* expression assessed by qPCR before ($-$) or after ($+$) αCD3/CD28 or PMA/ionomycin (P/I) stimulation of sorted $YFP^+$ $T_{reg}$ cells from WT and $Ndfip1^{fl/fl}$ $Foxp3$-Cre mice. A representative example of *Ndfip1* expression relative to *Actb* after αCD3/CD28 stimulation is shown. (**b**) Representative Haematoxylin and Eosin-stained histological sections of the skin, oesophagus and lung from genotypes as indicated are shown. Scale bars represent 100 μM. Far right image in panel is a representative image of the spleen and lymph nodes to illustrate size. (**c**) Inflammation index, calculated as a spleen weight/body weight for male $Ndfip1^{+/+}$ $Foxp3$-Cre (WT) and $Ndfip1^{fl/fl}$ $Foxp3$-Cre (cKO) mice at 9–16 weeks of age. (**d**) Levels of serum antibody isotypes as quantified by ELISA. (**e,f**) T cells from lung homogenates were analysed by flow cytometry for (**e**) the percentages of $CD44^+$ cells among $CD4^+$ cells and (**f**) the percentages of $CD4^+$ cells producing the indicated cytokines after *ex vivo* (P/I) stimulation. *P* values determined by Student's *t*-test, with correction for unequal variances as appropriate. *$P < 0.05$, **$P < 0.01$, ***$P < 0.001$, ****$P < 0.0001$. For **a,c–f**, bars indicate mean ± s.e.m. Data in **a** are representative of four male animals of each genotype, 9–16 weeks, and for **c–f** each dot represents an individual male mouse aged between 9 and 16 weeks. All experiments were performed on at least two independent occasions.

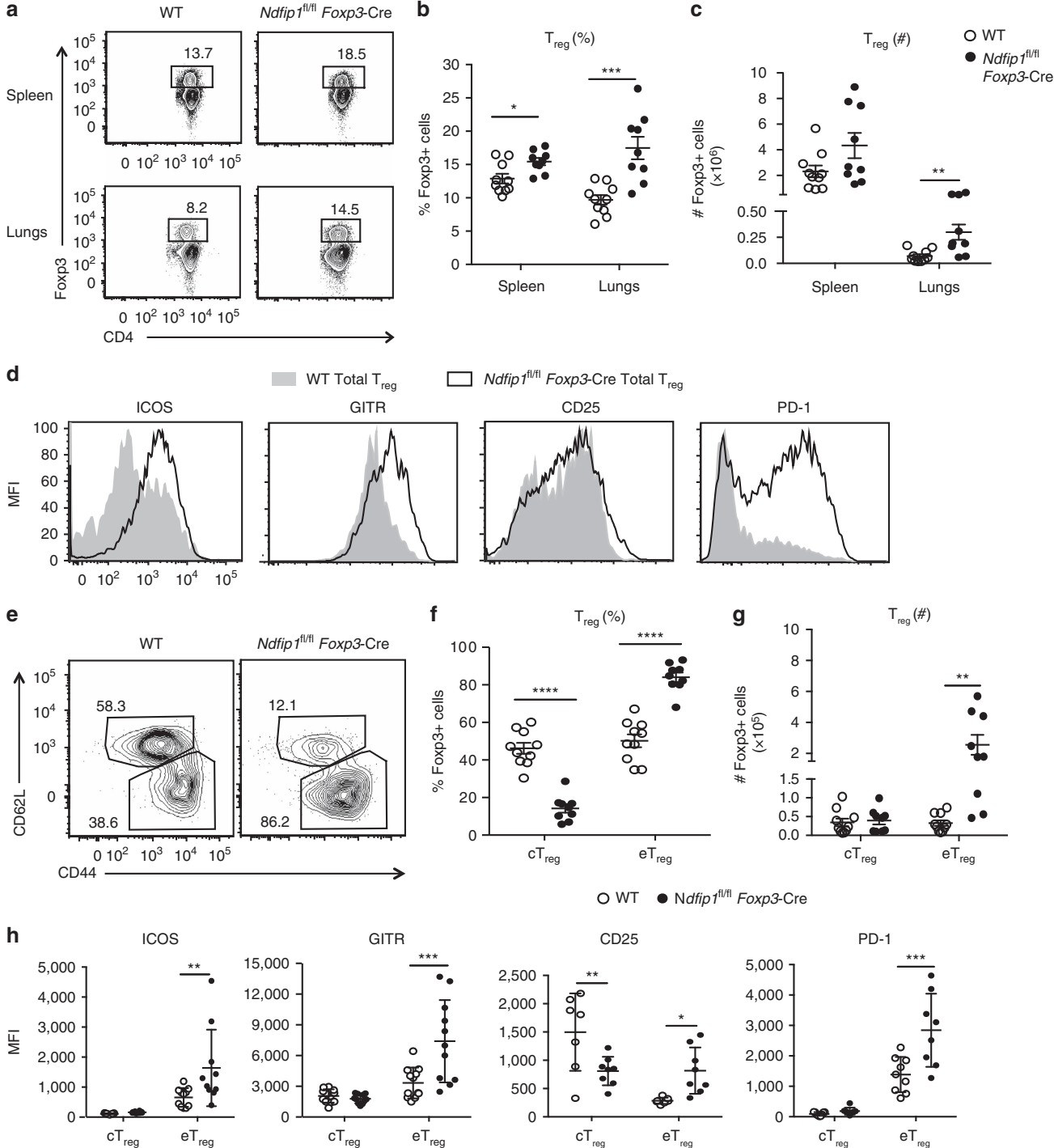

**Figure 2 | Ndfip1-deficient eT_reg cells are increased in number and T_reg cell surface expression.** T_reg cells from the spleens and lung (**a–c**) or lung (**d–h**) of 9–16-week-old male WT and *Ndfip1*^fl/fl^ *Foxp3*-Cre mice were analysed by flow cytometry for numbers and surface markers. (**a**) Representative flow plots of T_reg cells (previously gated as live CD3^+^CD4^+^ cells) from the spleen and lung homogenates that were used to determine the percentages and total numbers. (**b**) T_reg percentages and (**c**) numbers compiled over multiple experiments. (**d**) Representative histograms of lung T_reg cells (gated as in **a**) analysed for surface expression of ICOS, GITR, CD25 and PD-1. (**e**) Representative flow plots showing the gating of T_reg cells that are effector (eT_reg, CD44^hi^CD62L^lo^) or central (cT_reg, CD62L^hi^CD44^lo^). This gating was used to determine the (**f**) percentages and (**g**) total numbers of these cells. (**h**) Quantification of mean fluorescence intensities (MFIs) of ICOS, GITR, CD25 and PD-1 on cT_reg cells and eT_reg cells. *P* values were determined by Student's *t*-test, with correction for unequal variances as appropriate. *$P < 0.05$, **$P < 0.01$, ***$P < 0.001$, ****$P < 0.0001$. Each dot indicates data acquired from a single male mouse. Graphs with compiled data show mean ± s.e.m. All experiments were performed on at least two independent occasions.

Decreased suppressive function of T_reg cells is known to lead to inflammation. Therefore, we examined T_reg cell function. *In vitro*, Ndfip1-deficient and WT T_reg cells suppressed proliferation of WT T conventional (T_conv) cells to the same degree

(Supplementary Fig. 1c,d). We then examined T_reg cell function *in vivo* using a model of T-cell transfer-induced colitis. Similar to our results from the *in vitro* assays, Ndfip1-deficient and WT T_reg cells were equally able to prevent weight loss due to inflammation

caused by cotransferred $T_{conv}$ cells (Supplementary Fig. 1e). These data support that the pathology observed in $Ndfip1^{fl/fl}$ $Foxp3$-Cre mice is not due to a loss of $T_{reg}$ number or an overall loss of $T_{reg}$ cell-suppressive function.

**Ndfip1 limits $T_{reg}$ proliferation and levels of ICOS and GITR**. The expansion of $CD44^+$ $eT_{reg}$ cells in male $Ndfip1^{fl/fl}$ $Foxp3$-Cre animals in the absence of robust changes in $cT_{reg}$ cells suggested that Ndfip1 restricts $eT_{reg}$ cell numbers. Additionally, our data supported that Ndfip1 limits the expression of ICOS, GITR, CD25 and PD-1 on $eT_{reg}$ cells. However, both expansion and phenotype of $eT_{reg}$ cells could be altered in inflammatory environments, independent of an intrinsic role for Ndfip1. To distinguish between these possibilities, we examined $Ndfip1^{fl/fl}$ $Foxp3$-Cre$^{+/-}$ female animals. These mice have mixtures of WT (YFP-Cre$^-$) and Ndfip1-deficient (YFP-Cre$^+$) Foxp3$^+$ $T_{reg}$ cells, due to X chromosome inactivation, that can be distinguished by the expression of the YFP Cre reporter.

Surprisingly, despite the presence of WT $T_{reg}$ cells, female $Ndfip1^{fl/fl}$ $Foxp3$-Cre$^{+/-}$ mice had similar inflammatory burden, as defined by the ratio of spleen weight to body weight (Supplementary Fig. 2a) and contained equally high percentages of IFNγ-, IL-4- and IL-17A-expressing CD4 T cells, as the males (Supplementary Fig. 2b). Compared to male animals, which are uniformly sick beyond 16 weeks of age, female animals had lower incidence of skin dermatitis. $Ndfip1^{fl/fl}$ $Foxp3$-Cre$^{+/-}$ mice contained increased numbers of total lung $T_{reg}$ cells compared to $Ndfip1^{+/+}$ $Foxp3$-Cre$^{+/-}$ counterparts (Fig. 3a,b). Similar to their male counterparts, female animals had total lung $T_{reg}$ cells that were skewed towards the $eT_{reg}$ phenotype (Fig. 3c,d). In control $Ndfip1^{+/+}$ $Foxp3$-Cre$^{+/-}$ female mice, $Foxp3$-Cre$^+$ (YFP$^+$) cells were outnumbered by $Foxp3$-Cre$^-$ (YFP$^-$) cells by almost 1:3, indicating an effect of Cre expression (Fig. 3f). Surprisingly, in $Ndfip1^{fl/fl}$ $Foxp3$-Cre$^{+/-}$ females, this ratio was skewed towards YFP$^+$ cells, supporting an *in vivo* competitive advantage of Ndfip1-deficient $T_{reg}$ cells (Fig. 3f).

To identify intrinsic effects of Ndfip1 deficiency on $T_{reg}$ cells, in each female animal, for each parameter examined, we first determined the ratio of the value for YFP$^+$-to-YFP$^-$ $T_{reg}$ cells to normalize for an effect of Cre. Then we compared this ratio between $Ndfip1^{fl/fl}$ $Foxp3$-Cre$^{+/-}$ and $Ndfip1^{+/+}$ $Foxp3$-Cre$^{+/-}$ animals. Our analysis revealed that Ndfip1-deficient $T_{reg}$ cells were significantly more likely to display an $eT_{reg}$ phenotype (CD44$^+$CD62L$^{lo}$) (Fig. 3g). Furthermore, expression of the proliferative marker Ki67 (Fig. 3h) was increased in Ndfip1-deficient $eT_{reg}$ cells. $eT_{reg}$ cells lacking Ndfip1 also had significantly higher levels of ICOS (Fig. 3i) and GITR (Fig. 3j). Thus Ndfip1 restricts $eT_{reg}$ numbers by limiting their proliferation and their expression of ICOS and GITR. However, levels of CD25 and PD-1 were similar to controls (Fig. 3k,l), suggesting that these markers were not directly affected by the loss of Ndfip1 in $T_{reg}$ cells.

The expression of activation markers on $T_{reg}$ cells is known to increase under inflammatory settings[20]. To determine how inflammatory conditions could influence control $T_{reg}$ cells, we looked at the expression of surface markers on CD44$^+$ YFP$^-$ $T_{reg}$ cells from the $Ndfip1^{+/+}$ $Foxp3$-Cre$^{+/-}$ females or from the inflamed environment in $Ndfip1^{fl/fl}$ $Foxp3$-Cre$^{+/-}$ female mice. CD44$^+$ YFP$^-$ $T_{reg}$ cells from $Ndfip1^{fl/fl}$ $Foxp3$-Cre$^{+/-}$ mice showed increased expression of ICOS, GITR, CD25 and PD-1 (Supplementary Fig. 2c–f). Taken together with the data in Figs 2 and 3, this supports that the changes in CD25 and PD-1 expression in male $Ndfip1^{fl/fl}$ $Foxp3$-Cre mice are secondary to inflammation and that while inflammation can increase levels of ICOS and GITR, Ndfip1 is also an intrinsic regulator of ICOS and GITR on $T_{reg}$ cells (Supplementary Fig. 2g).

To further address whether the changes observed in the Ndfip1-deficient $T_{reg}$ cells were driven by inflammation, we looked in neonatal mice before the onset of overt inflammation. In 13-day-old female neonates, we found no differences between $Ndfip1^{fl/fl}$ $Foxp3$-Cre$^{+/-}$ and $Ndfip1^{+/+}$ $Foxp3$-Cre$^{+/-}$ animals in spleen weight (Supplementary Fig. 3a), inflammation index (Supplementary Fig. 3b), total lung Foxp3$^+$ $T_{reg}$ cell number or cytokine-producing CD4 T cells (Supplementary Fig. 3c,d). However, lungs from $Ndfip1^{fl/fl}$ $Foxp3$-Cre$^{+/-}$ female mice contained greater frequencies of YFP$^+$ $T_{reg}$ cells (Supplementary Fig. 3e) and greater frequencies of $eT_{reg}$ cells (Supplementary Fig. 3f). Further, these $eT_{reg}$ cells showed higher expression of ICOS and GITR (Supplementary Fig. 3g,h). This further supports that Ndfip1 limits $eT_{reg}$ cell frequency and expression of ICOS and GITR in an intrinsic manner.

**Ndfip1 limits IL-4 production by $T_{reg}$ cells**. Our data indicate that Ndfip1 limits $eT_{reg}$ cell expression of ICOS and GITR and restricts $eT_{reg}$ cell proliferation. However, it remained unclear why $Ndfip1^{fl/fl}$ $Foxp3$-Cre$^{+/-}$ females developed inflammation, when approximately half of the $T_{reg}$ cells in these mice are Ndfip1 sufficient. This suggested a pathological gain of function in Ndfip1-deficient $T_{reg}$ cells. We thus investigated whether Ndfip1-deficient $T_{reg}$ cells could contribute to the pool of cytokine-producing cells in the lung of $Ndfip1^{fl/fl}$ $Foxp3$-Cre male mice shown in Fig. 1. Effector cytokine production by WT $T_{reg}$ cells is relatively rare but has been described under inflammatory[21] or lymphopenic settings[22,23]. Strikingly, while WT $T_{reg}$ cells did not produce any IL-4 upon *ex vivo* stimulation, Ndfip1-deficient $T_{reg}$ cells could produce IL-4, and this was detectable at both the protein (Fig. 4a,b) and mRNA levels (Fig. 4c). Ndfip1-deficient $T_{reg}$ cells were also more likely to produce IL-10, IFNγ and IL-17A relative to WT controls. To determine whether the cytokine production was due to the loss of Ndfip1 or tied to the inflammatory environment, we generated mixed bone marrow chimeras. Consistent with our data from the female mice, these mixed chimeras developed dermatitis 8 weeks after reconstitution. Upon restimulation of lung homogenate, CD45.1$^+$ WT Foxp3$^+$ cells did not produce any IL-4 while CD45.2$^+$ Ndfip1-deficient Foxp3$^+$ cells in the same host could produce IL-4 (Fig. 4d,e), supporting an intrinsic role for Ndfip1 in limiting IL-4 production from $T_{reg}$ cells. Surprisingly, when we analysed the Foxp3$^-$ $T_{conv}$ cells, we found a significant population of CD45.2$^+$ cells that also expressed IL-4 (Fig. 4f,g), raising the possibility that the IL-4-producing population originated from $Ndfip1^{fl/fl}$ $Foxp3$-Cre $T_{reg}$ cells that had lost Foxp3.

**Ndfip1-deficient $T_{reg}$ cells lose Foxp3 expression**. Given the IL-4 production by Ndfip1-deficient $T_{reg}$ cells and their increased proliferative capacity, we posited that these cells would be likely to become methylated at their $Foxp3$ locus and become unstable. IL-4 receptor signalling in $T_{reg}$ cells, via signal transducer and activator of transcription factor 6 (STAT6), results in the methylation of the CNS2 region of the $Foxp3$ locus and repression of $Foxp3$ gene expression[24]. Therefore, we compared methylation at 12 Cytosine-phosphate-guanine (CpG) islands in the $Foxp3$ CNS2 region in Ndfip1-sufficient and -deficient $T_{reg}$ cells, as well as WT $T_{conv}$ cells. As expected, the CNS2 CpG motifs were predominantly methylated in $T_{conv}$ cells (Supplementary Fig. 4a)[24]. Both WT and Ndfip1-deficient $T_{reg}$ cells had unmethylated CpG motifs in their $Foxp3$ promoter regions (Supplementary Fig. 4b), consistent with their expression of $Foxp3$ mRNA (Supplementary Fig. 4c). However, in $Ndfip1^{fl/fl}$ $Foxp3$-Cre $T_{reg}$ cells, there was an increase in methylation at the

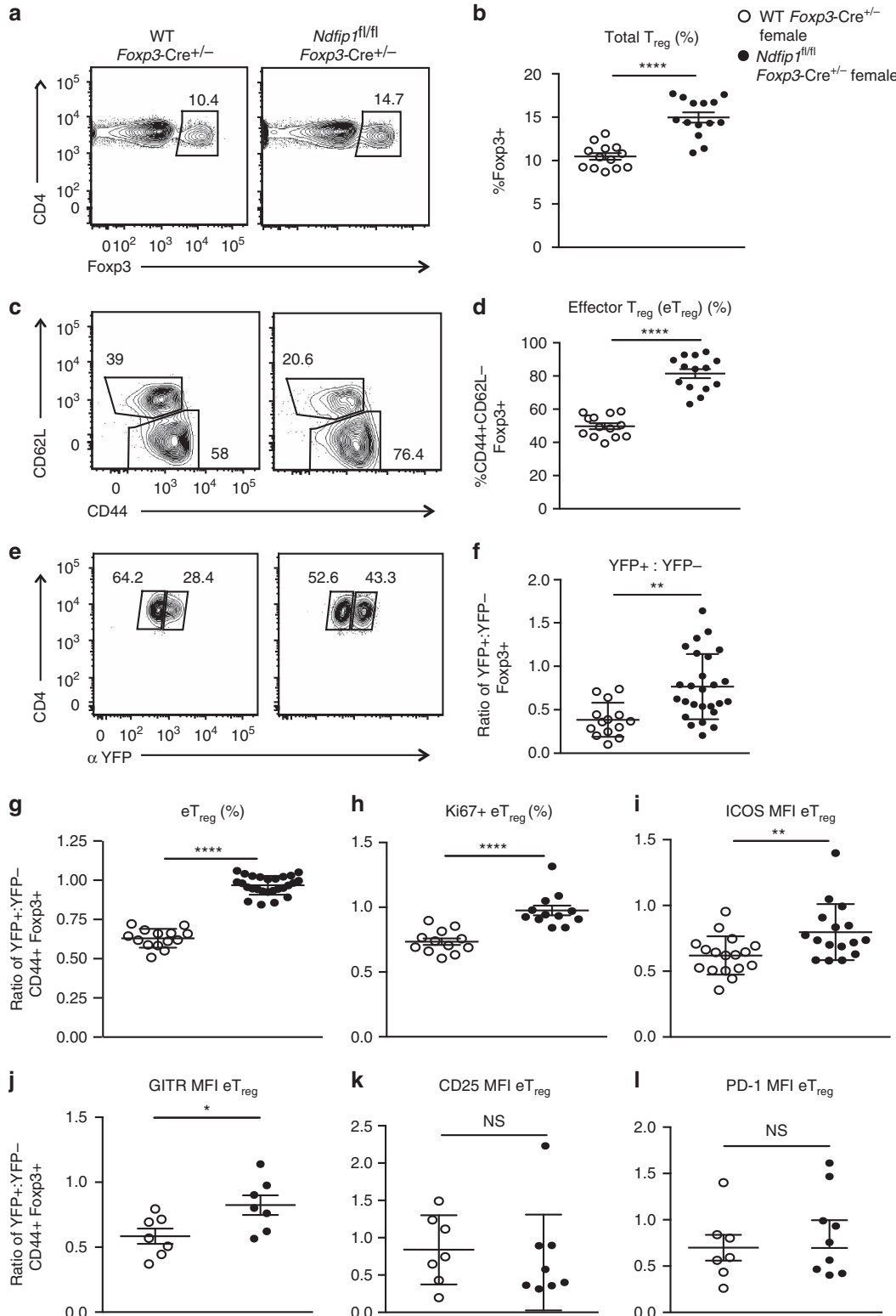

**Figure 3 | Female *Ndfip1*^fl/fl^*Foxp3*-Cre^+/−^ mice show similar T_reg cell changes as males.** (a–l) Lung homogenates, from 8- to 16-week-old hemizygous female *Ndfip1*^fl/fl^ *Foxp3*-Cre^+/−^ mice and *Ndfip1*^+/+^ *Foxp3*-Cre^+/−^ controls, were analysed *ex vivo* by flow cytometry. (**a**) Representative flow plots showing the gating of T_reg cells (previously gated as live CD3^+^CD4^+^ cells). This gating was used to determine the overall frequencies of T_reg cells (**b**) in mice from the indicated genotypes. (**c**) Representative flow plots showing the gating of T_reg cells from **a** that are eT_reg or cT_reg cells. This gating was used to determine the (**d**) frequencies of eT_reg cells. (**e**) Representative flow plot of T_reg cells as gated in **a** and analysed for frequencies of YFP^+^ and YFP^−^ cells. (**f**) Ratio of YFP^+^ to YFP^−^ T_reg cells. (**g,h**) Percentages of (**g**) eT_reg cells or (**h**) Ki-67^+^ eT_reg cells, shown as a ratio of YFP^+^:YFP^−^ cells within each mouse to normalize for a Cre effect. (**i–l**) mean fluorescence intensity (MFI) of (**i**) ICOS, (**j**) GITR, (**k**) CD25 and (**l**) PD-1 expression on eT_reg cells, again shown as a ratio of YFP^+^:YFP^−^. *P* values determined by Student's *t*-test, with correction for unequal variances as appropriate. *$P<0.05$, **$P<0.01$, ***$P<0.001$, ****$P<0.0001$. Each dot shows data acquired from a single female mouse. Graphs show mean ± s.e.m. All experiments were performed on at least two independent occasions.

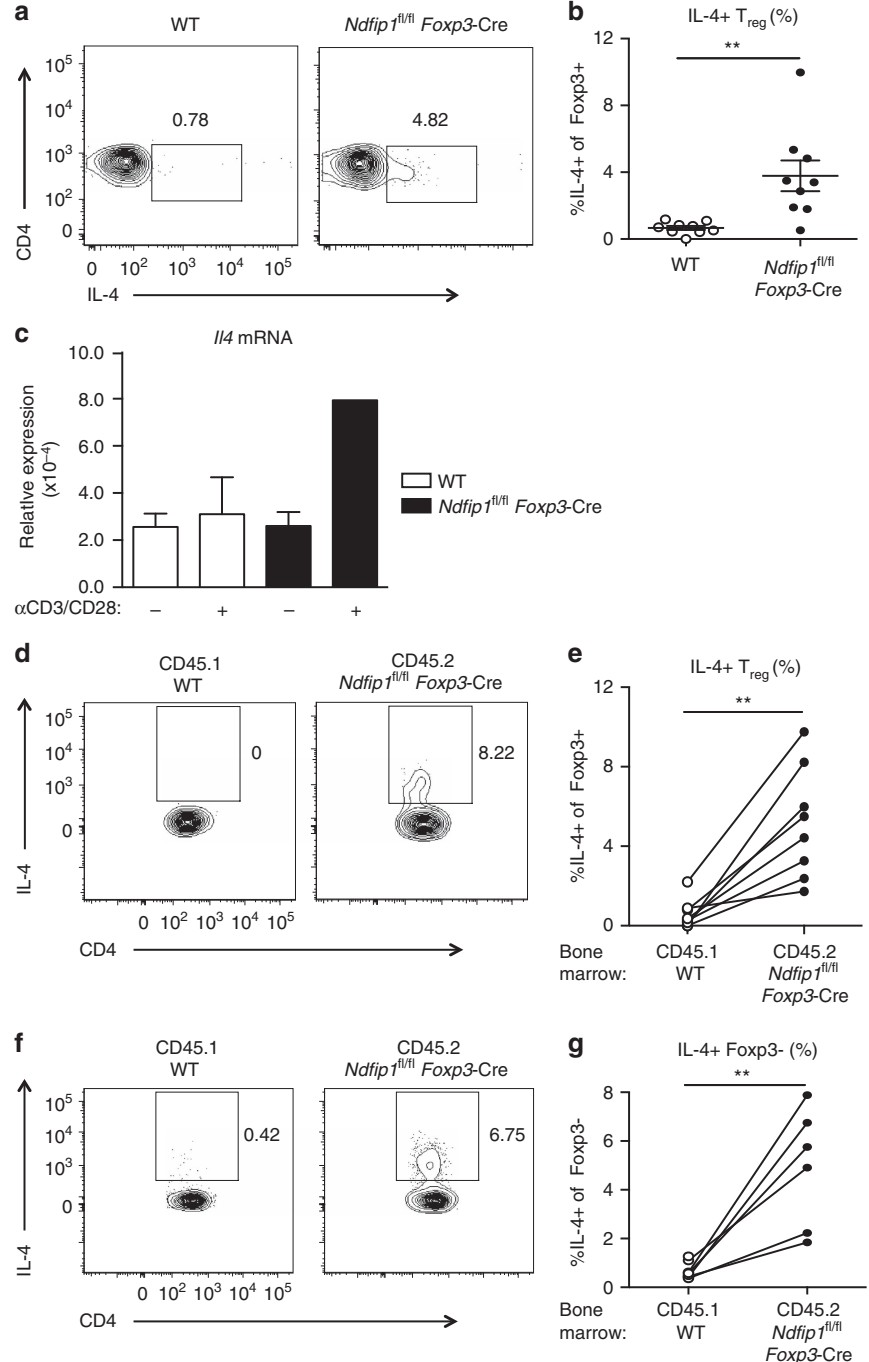

**Figure 4 | Ndfip1-deficient T$_{reg}$ cells make IL-4.** (**a,b**) Lung homogenates from WT and *Ndfip1*$^{fl/fl}$ *Foxp3*-Cre male mice aged 9–16 weeks were stimulated *ex vivo* with P/I and analysed for IL-4 production by flow cytometry. (**a**) Representative flow plots of T$_{reg}$ cells (Foxp3$^+$CD3$^+$CD4$^+$) producing IL-4. (**b**) Compiled percentages of IL-4-producing T$_{reg}$ cells. (**c**) qPCR analysis of *Il4* from sorted YFP$^+$ T$_{reg}$ cells from WT and *Ndfip1*$^{fl/fl}$ *Foxp3*-Cre male mice. Cells were unstimulated ( − ) or stimulated ( + ) with αCD3/CD28. *Il4* mRNA is shown relative to *Actb*. (**d–g**) Lung homogenates from mixed chimeras were stimulated *ex vivo* with P/I and analysed for IL-4 production by flow cytometry. (**d**) Representative flow plots showing IL-4 production from T$_{reg}$ cells from WT (gated on CD45.1) or *Ndfip1*$^{fl/fl}$ *Foxp3*-Cre (gated on CD45.2) cells from the same recipient. (**e**) Graphs showing the percentages of IL-4$^+$ T$_{reg}$ cells from CD45.1 and CD45.2 cells in each chimera; cells from the same recipient are connected by a line. (**f**) Representative flow plots showing IL-4 production from T$_{conv}$ cells from the same chimeras. (**g**) Graphs showing the percentages of IL-4$^+$ T$_{conv}$ cells as per **e**. All experiments were performed on at least two independent occasions. Graphs show mean ± s.e.m. Mixed bone marrow chimeras were generated and analysed in two separate experiments using male donors. *P* values were determined by Student's *t*-test for **b,c** or paired *T*-test. *$P < 0.05$, **$P < 0.01$.

12 examined CNS2 CpG sites, compared to WT T$_{reg}$ cells, which remained unmethylated at these sites (Fig. 5a,b).

These data suggested that Ndfip1 deficiency, via concomitant expression of IL-4, negatively impacts lineage stability of T$_{reg}$ cells. To test whether Ndfip1-sufficient and -deficient T$_{reg}$ cells have differential responses to destabilizing cytokines, we cultured Ndfip1-sufficient and -deficient Foxp3$^+$ cells in stabilizing (IL-2) or destabilizing (IL-4 plus anti-IL-2) conditions and analysed Foxp3 protein expression as a surrogate for lineage stability. We found that Ndfip1-deficient T$_{reg}$ cells were equally stabilized

by IL-2 and destabilized by IL-4 compared to their Ndfip1-sufficient counterparts (Supplementary Fig. 4d,e). Thus Ndfip1-deficient $T_{reg}$ cells are not uniquely sensitive to the

destabilizing effects of IL-4; *in vivo*, however, Ndfip1-deficient IL-4 producing $T_{reg}$ cells are more likely to encounter destabilizing cytokine milieus.

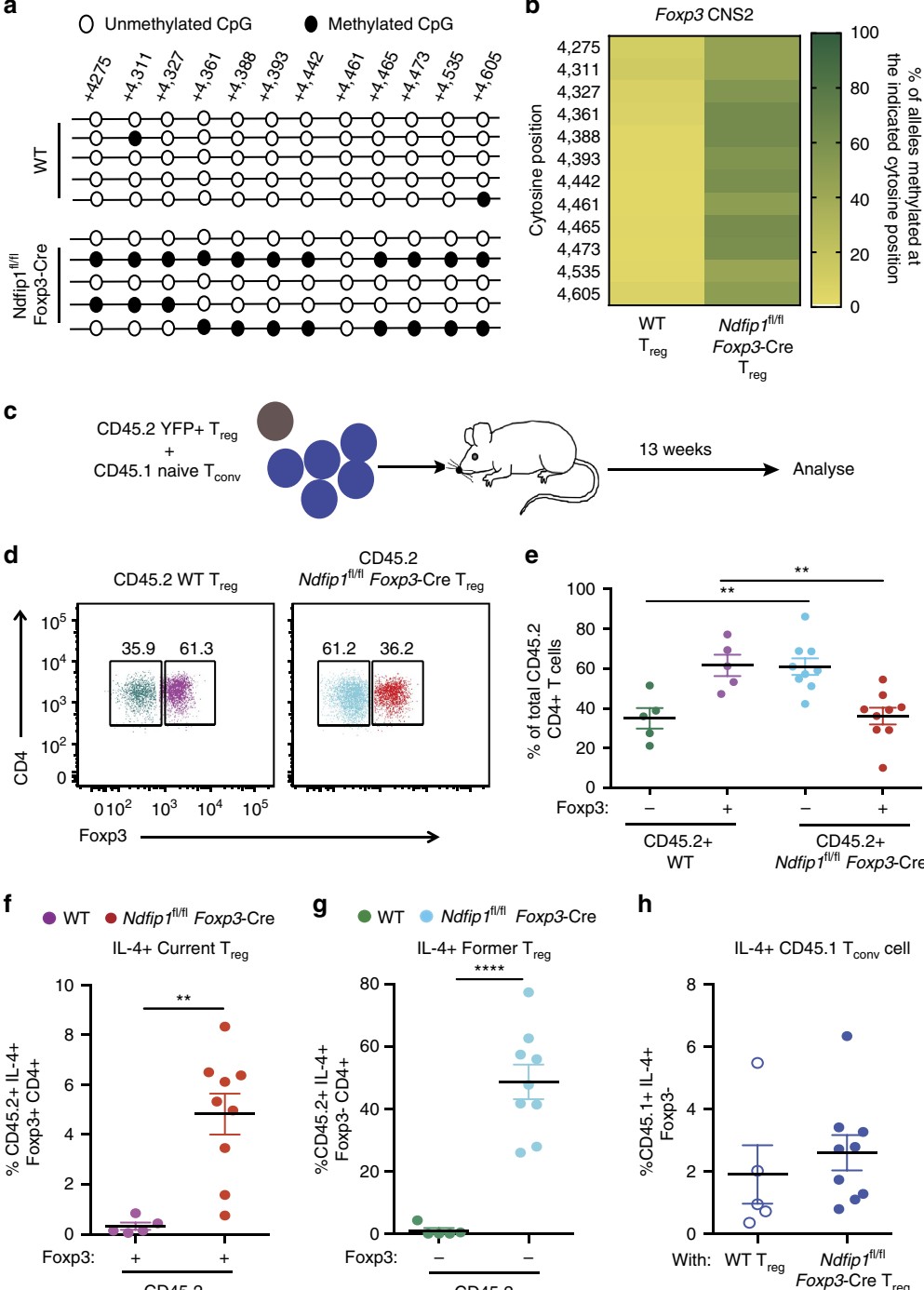

**Figure 5 | Ndfip1-deficient $T_{reg}$ cells have *Foxp3* locus instability and lose Foxp3 *in vivo*.** (**a,b**) YFP$^+$ e$T_{reg}$ cells were sorted from 9- to 12-week-old WT or *Ndfip1*$^{fl/fl}$ *Foxp3*-Cre male mice and assessed for CNS2 methylation using bisulfite sequencing. (**a**) Representative data of methylation at the *Foxp3* CNS2 locus in $T_{reg}$ cells from the two genotypes. (**b**) Quantification of methylation at 12 CpG islands in the *Foxp3* CNS2 locus. Data shown are compiled from three mice of each genotype. Data are the percentage of alleles methylated at each position. (**c–h**) YFP$^+$ $T_{reg}$ cells from CD45.2$^+$ WT or *Ndfip1*$^{fl/fl}$ *Foxp3*-Cre mice were mixed 1:5 with CD45.1$^+$ WT naive $T_{conv}$ cells and transferred to *Rag1*$^{-/-}$ recipients. Cells from lung homogenates were analysed for Foxp3 and IL-4 expression using flow cytometry 13 weeks after transfer. (**d**) A representative flow plot showing CD45.2$^+$ WT and *Ndfip1*$^{fl/fl}$ *Foxp3*-Cre cells that were analysed for the expression of Foxp3. (**e**) Compiled data from multiple mice analysed as in **d**. (**f**) Percentages of IL-4-producing cells that remained Foxp3$^+$ (current $T_{reg}$ cells ) and (**g**) that had become Foxp3$^-$ following transfer (former $T_{reg}$ cells ). (**h**) Percentages of IL-4-producing CD45.1 $T_{conv}$ cells. All experiments were performed on at least two separate occasions. Each dot represents cells from an individual mouse. *P* values were calculated by one-way ANOVA. *$P < 0.05$, **$P < 0.01$, ***$P < 0.001$, ****$P < 0.0001$. Graphs show mean ± s.e.m.

To test $T_{reg}$ lineage stability *in vivo*, we sorted WT or Ndfip1-deficient CD45.2$^+$ YFP$^+$ $T_{reg}$ cells and cotransferred them with sorted naive CD45.1$^+$ WT $T_{conv}$ cells into $Rag1^{-/-}$ hosts (Fig. 5c). As we had previously observed (Supplementary Fig. 1e), mice receiving either Ndfip1-deficient or -sufficient $T_{reg}$ cells did not develop colitis (Supplementary Fig. 5a). At harvest, we determined the frequencies of CD45.2$^+$ cells that were still Foxp3$^+$ (current $T_{reg}$ cells) or that had lost Foxp3 (former $T_{reg}$ cells). In mice that received WT $T_{reg}$ cells, on average, 30% had lost Foxp3, similar to previous reports by other groups[8,22]. Strikingly, in mice that had received Ndfip1-deficient $T_{reg}$ cells, 60% on average had lost expression of Foxp3 (Fig. 5d,e). These Foxp3$^-$ CD45.2$^+$ cells could represent Ndfip1-deficient $T_{reg}$ cells that had lost Foxp3 expression *in vivo* or a very proliferative $T_{conv}$ cell contamination from cell sorting. If the Foxp3$^-$ cells represented former $T_{reg}$ cells, they would lack *Ndfip1* mRNA, due to excision following the expression of Foxp3-Cre. Contaminating $T_{conv}$ cells, on the other hand, would continue to express *Ndfip1*. We sorted current $T_{reg}$ cells, former $T_{reg}$ cells and $T_{conv}$ cells and stimulated to induce the *Ndfip1* mRNA expression. We found that the WT $T_{conv}$ cells expressed *Ndfip1* mRNA, while neither the Ndfip1$^{fl/fl}$*Foxp3-Cre*$^+$ current $T_{reg}$ cells nor former $T_{reg}$ cells expressed detectable levels of *Ndfip1* mRNA (Supplementary Fig. 5b). Thus $T_{reg}$ cells lacking Ndfip1 were more likely to lose Foxp3 expression than their WT counterparts.

Current (Fig. 5f) and former (Fig. 5g) $T_{reg}$ cells lacking Ndfip1 produced IL-4, while current and former WT $T_{reg}$ cells did not. Importantly, none of the WT $T_{conv}$ cells produced IL-4 (Fig. 5h). This was not true for all cytokines since WT and Ndfip1-deficient cells were equally likely to make IFN$\gamma$ regardless of whether they were $T_{reg}$ cells (Supplementary Fig. 5c,d) or cotransferred $T_{conv}$ cells (Supplementary Fig. 5e).

Additionally, compared to WT $T_{reg}$ cells, Ndfip1-deficient $T_{reg}$ cells expanded to greater numbers *in vivo* (Supplementary Fig. 5f,g) suggesting an advantage in proliferation and/or survival of both current and former Ndfip1-deficient $T_{reg}$ cells. However, at steady state, it is likely that only a small percentage of $Ndfip1^{fl/fl}Foxp3-Cre^+$ $T_{reg}$ cells are actively losing their Foxp3 at any point of time. In support of this, Ndfip1-deficient total $T_{reg}$ cells, $eT_{reg}$ cells and $cT_{reg}$ cells from female animals do not show a decrease in Foxp3 mean fluorescence intensity at steady state (Supplementary Fig. 6).

In summary, in a setting where $T_{reg}$ cells are pushed to undergo lymphopenia-induced proliferation, Ndfip1-deficient $T_{reg}$ cells have an advantage in growth and expansion, which leads to a large increase in total numbers of Ndfip1-deficient current and former $T_{reg}$ cells. Since the majority of Ndfip1-deficient former $T_{reg}$ cells ($\sim$50%) produce IL-4, this may explain the dramatic loss of Foxp3 observed *in vivo*.

**IL-4 is dispensable for Ndfip1-deficient $eT_{reg}$ expansion.** To assess whether IL-4 is required for the phenotypic changes in $T_{reg}$ cells lacking Ndfip1, we examined $T_{reg}$ cells in mice that lack both *Ndfip1* and *IL-4* (*Ndfip1 IL-4* double knockout or *DKO* mice)[25]. These animals do not show the overt signs of inflammation such as dermatitis[25,26] observed in age-matched $Ndfip1^{fl/fl}$ *Foxp3-Cre* animals. We observed increased frequencies of $T_{reg}$ cells (Supplementary Fig. 7a,b) compared to IL-4-deficient controls. The Ndfip1-deficient $T_{reg}$ cells that lacked IL-4 were more proliferative than controls, as determined by Ki67 (Supplementary Fig. 7c). Similar to the $Ndfip1^{fl/fl}$ *Foxp3-Cre* mice, $T_{reg}$ cells in the *Ndfip1 IL-4 DKO* mice were predominantly $eT_{reg}$ cells (Supplementary Fig. 7d,e). Additionally, the Ndfip1- and IL-4-deficient $eT_{reg}$ cells expressed higher levels of Ki67 (Supplementary Fig. 7f) and ICOS (Supplementary Fig. 7g)

relative to control $eT_{reg}$ cells. Thus, while IL-4 from current and former $T_{reg}$ cells likely contributes to the inflammation observed in $Ndfip1^{fl/fl}$ *Foxp3-Cre* mice, increased $T_{reg}$ cell activation, proliferation and ICOS expression are not due solely to IL-4 signalling. Thus IL-4 is insufficient to explain the increased fitness of Ndfip1-deficient $T_{reg}$ cells.

**Ndfip1-deficient $eT_{reg}$ cells have altered metabolic activity.** To identify molecular pathways underlying the increased proliferation and altered $eT_{reg}$ cell phenotype of Ndfip1 deficient $T_{reg}$ cells, we used label-free quantitative proteomics to compare WT and Ndfip1-deficient $cT_{reg}$ and $eT_{reg}$ cells. We sorted YFP$^+$ $eT_{reg}$ and $cT_{reg}$ cells from young WT or $Ndfip1^{fl/fl}$ *Foxp3-Cre* male mice, subjected these cells to liquid chromatography-tandem mass spectrometry (LC-MS/MS) analysis and quantified protein abundance using intensity-based absolute quantification[27–29] (Fig. 6a). We achieved good reproducibility of identified proteins across three experiments, pooling two to three mice of each genotype per experiment (Fig. 6b). Comparing proteins identified in control $eT_{reg}$ and $cT_{reg}$ proteomes, our data fit well with what has been published previously on proteins unique to each subtype. Namely, among proteins that were significantly more abundant in $cT_{reg}$ than $eT_{reg}$ proteomes or that were found exclusively in $cT_{reg}$ proteomes, defined here as having a $cT_{reg}$ 'bias', we identified CD62L and Bcl2 (ref. 20) (Fig. 6c, Supplementary Data 1). Proteins that were increased in, or biased toward, the $eT_{reg}$ proteome included CD44 and several proteins associated with $eT_{reg}$ cells, such as Integrin alpha E/itgae (CD103)[30] (Fig. 6c, Supplementary Data 1).

To characterize differential protein expression in Ndfip1-sufficient and -deficient $T_{reg}$ cells, we compared the ratio of protein abundance in $eT_{reg}$:$cT_{reg}$ for each of the three experiments. This revealed widespread changes in the proteomes of Ndfip1-deficient $T_{reg}$ cells. Given that Ndfip1 deficiency drove changes predominantly in $eT_{reg}$ cells, we focussed our comparisons on the levels of proteins identified in WT and Ndfip1-deficient $eT_{reg}$ cells (Supplementary Data 2 and Fig. 6d). Our flow cytometric finding of increased GITR expression was supported by these $T_{reg}$ proteomes (Supplementary Data 2 and Fig. 6d). Surprisingly, proteins associated with increased mTORC1 activity were increased in Ndfip1-deficient $eT_{reg}$ cells. This included Lamtor1 (Fig. 6d), as well as the V-ATPase subunit d1 and Lamtor3, which are components of the v-ATPase-regulator complex that drives mTORC1 activation in cells[31–33]. We then performed network visualization of gene-ontology (GO) enrichment analysis for differentially regulated proteins in Ndfip1-sufficient and -deficient $eT_{reg}$ cells and identified clusters of nodes relating to metabolic processes that were significantly enriched (Fig. 6e and Supplementary Data 2). Pathway analysis further supported changes in the metabolic state of Ndfip1-deficient $eT_{reg}$ cells and, more specifically, an increase in mTORC1 activity. Thus loss of Ndfip1 led to an altered proteomic profile indicative of altered metabolic activity.

**Ndfip1-deficient $T_{reg}$ cells have increased glycolysis.** Based on this proteomics profiling, we sought to analyse the metabolic capacity of Ndfip1-deficient $T_{reg}$ cells. To obtain sufficient cell numbers for metabolic testing, we used *in vitro* IL-2-expanded $T_{reg}$ cells. We evaluated the bioenergetics of these expanded WT and $Ndfip1^{fl/fl}$ *Foxp3-Cre* $T_{reg}$ cells at 'rest' and also upon restimulation. We measured the extracellular acidification rate under glycolytic stress conditions (Fig. 7a–d) and oxygen consumption rate under mitochondrial stress conditions (Fig. 7e–h). Prior to stimulation, WT $T_{reg}$ cells had low basal rates

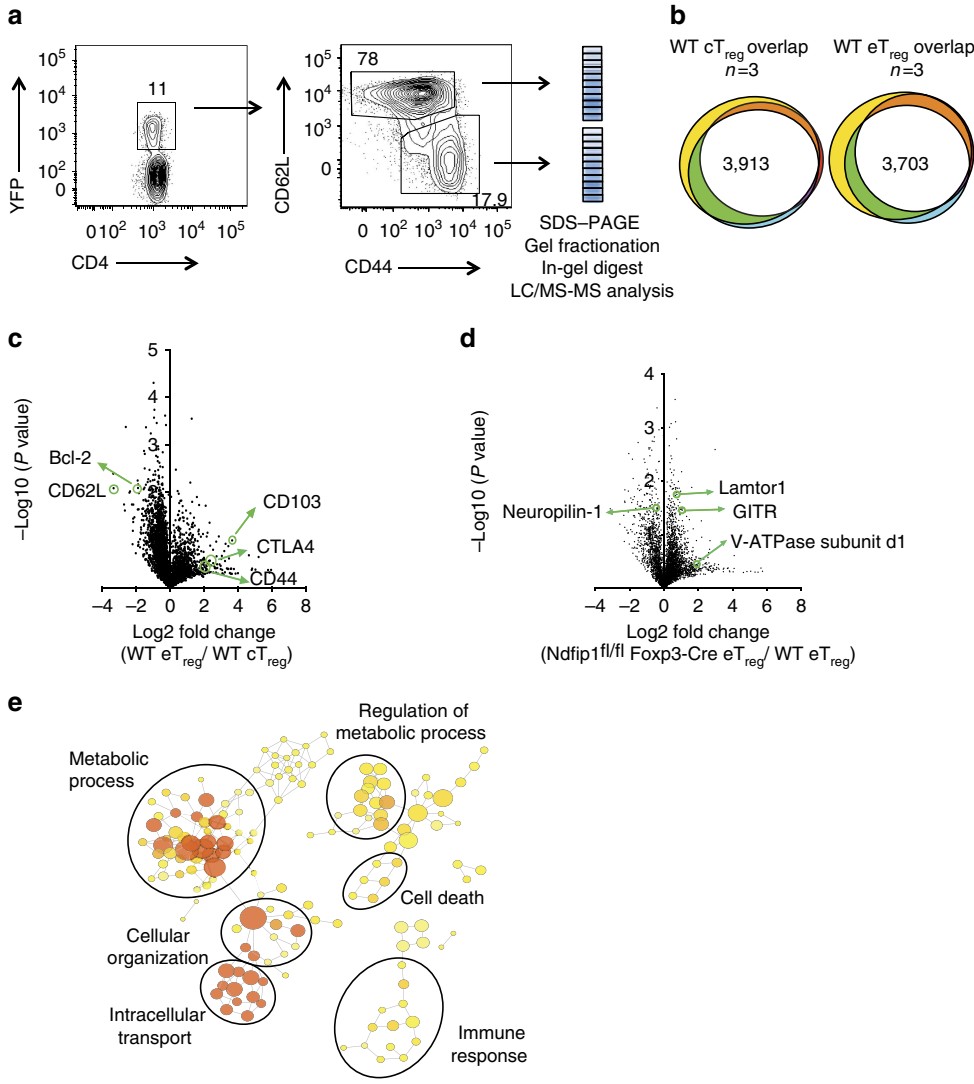

**Figure 6 | *Ndfip1*[fl/fl]*Foxp3*-Cre eT_reg cells have altered metabolic whole-cell proteome.** (**a**) Representative flow cytometry dot plots and gel pixelation to illustrate procedure used to analyse cT_reg and eT_reg cell proteomes using mass spectrometry. (**b**) Area-proportional Venn diagrams illustrating the reproducibility of proteins identified in cT_reg (left) or eT_reg (right) cells from three independent experiments. WT cells were compared for this analysis and data were compiled using the intensity-based absolute quantification method. (**c**) A volcano plot illustrating differentially expressed proteins between eT_reg versus cT_reg cells in WT mice. (**d**) A volcano plot representing differentially expressed proteins between eT_reg cells from *Ndfip1*[fl/fl] *Foxp3*-Cre or WT animals. (**e**) Network diagram of enriched GO terms with nodes representing GO annotations and edges connecting similar terms based on the GO hierarchy. The size of the nodes corresponds to the number of genes associated with the respective GO term and the colour of the nodes corresponds to the level of significance of the enrichment of the respective term in the data set (darker colour corresponds to a higher level of significance). Clusters of GO terms were manually analysed and annotated to identify broad functional similarity. Data are shown for YFP[+] eT_reg cells that were sorted from 9- to 12-week-old WT or *Ndfip1*[fl/fl] *Foxp3-Cre* male mice.

of glycolysis, but rates increased when cells were stimulated (Fig. 7a,b). In Ndfip1-deficient T_reg cells, the 'resting' glycolytic rate was only modestly increased over WT cells (Fig. 7a,c). However, restimulated Ndfip1-deficient T_reg cells had a considerably increased glycolytic rate and glycolytic capacity compared to restimulated WT T_reg cells (Fig. 7b–d). To assess mitochondrial function, we measured cell respiratory control, which is a general test of mitochondrial function in cells. Loss of Ndfip1 did not change the maximum respiratory capacity of T_reg cells and caused a slight decrease in the spare respiratory capacity of T_reg cells (Fig. 7e–h). Taken together, these data indicate that the loss of Ndfip1 drives a metabolic switch in activated T_reg cells, promoting a more effector cell-like reliance on glycolysis to supply cellular energetic demands.

**Ndfip1-deficient T_reg cells have elevated mTORC1 activity.** Effector T cells utilize glycolysis while T_reg cells are more dependent on oxidative phosphorylation as their main source of energy[34–38]. High glycolytic activity in T_reg cells, as can occur when mTORC1 activity is increased, has been associated with T_reg cell dysfunction[6,7,39]. To investigate whether mTORC1 activity was increased in Ndfip1-deficient T_reg cells, we cultured WT and *Ndfip1*[fl/fl] *Foxp3*-Cre T_reg cells *in vitro*. We found that *Ndfip1*[fl/fl] *Foxp3*-Cre T_reg cells quickly outnumbered their WT counterparts (Fig. 8a), were increased in cell size (Fig. 8b) and proliferated more (Fig. 8c). This, together with observed increases in ICOS and GITR expression (Fig. 8d,e), suggested that this *in vitro* culture system recapitulates the phenotype of Ndfip1-deficient T_reg cells *in vivo*. Given that ICOS can be driven by mTORC1 (ref. 39)

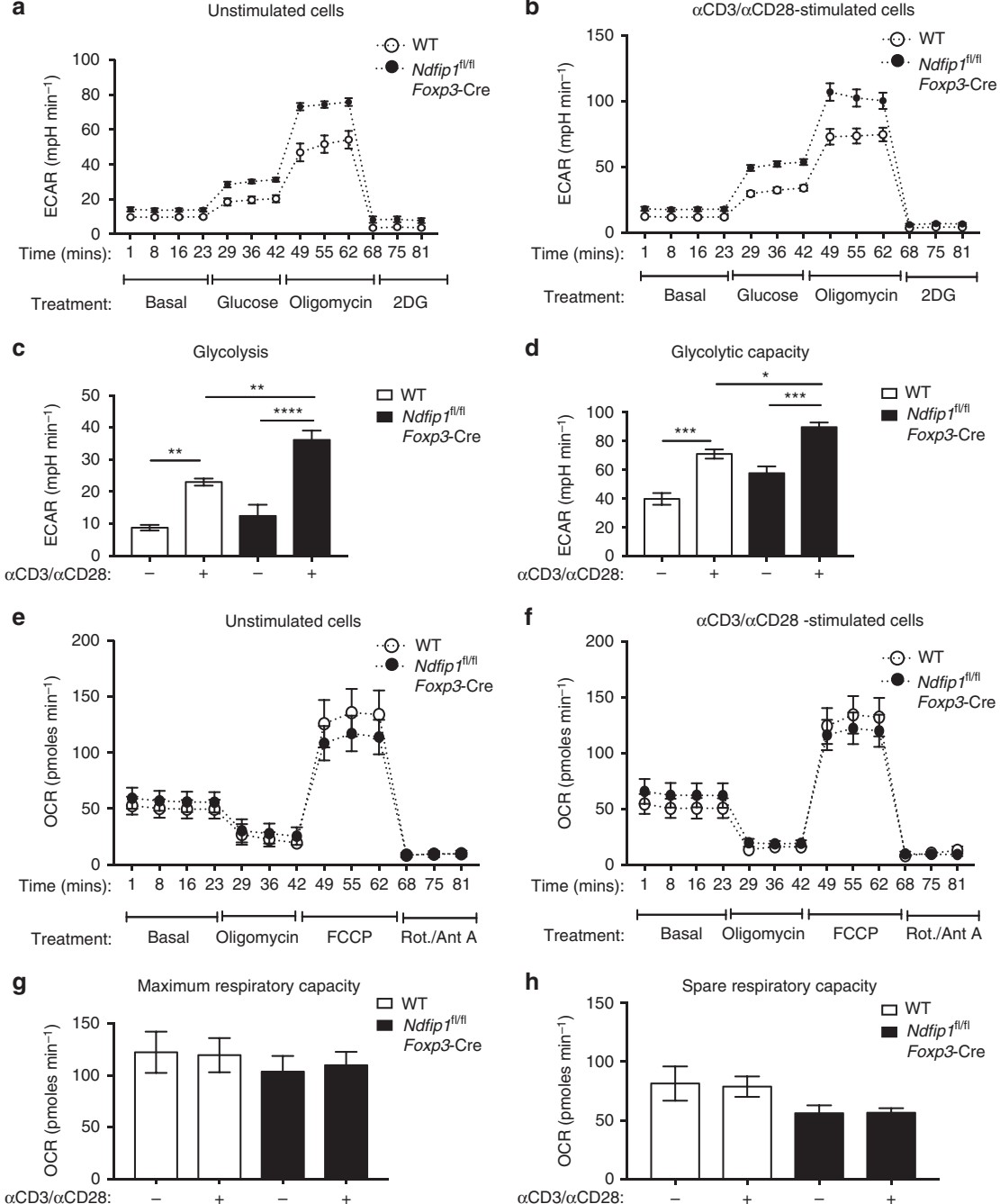

**Figure 7 | Ndfip1-deficient $T_{reg}$ cells have a significantly increased rate of glycolysis.** (**a–h**) YFP$^+$ $T_{reg}$ cells were sorted from *Ndfip1$^{+/+}$ Foxp3*-Cre WT or *Ndfip1$^{fl/fl}$ Foxp3-Cre* mice, expanded in culture and then were left unstimulated or were restimulated before metabolic function was assessed. (**a,b**) Extracellular acidification rate (ECAR) was measured during a glycolysis stress test in **a** unstimulated or (**b**) stimulated $T_{reg}$ cells treated with drugs as indicated. (**c**) Rate of glycolysis is the difference in ECAR between post-glucose addition and baseline. (**d**) Glycolytic capacity is the difference between post-Oligomycin ECAR and baseline ECAR. (**e,f**) Oxygen consumption rate (OCR) changes during a mitochondrial function assessment test in **e** unstimulated or (**f**) stimulated WT or *Ndfip1$^{fl/fl}$ Foxp3*-Cre $T_{reg}$ cells treated with drugs as indicated. (**g**) Maximum respiratory capacity is the difference in OCR after addition of rotenone/Antimycin A versus after addition of fluoro-carbonyl cyanide phenylhydrazone (FCCP). (**h**) Spare respiratory capacity is the difference in increased OCR following addition of FCCP compared to baseline. Graphs show mean ± s.e.m. (**a–d**) Represents $n = 4$ mice (male or female, 7–12 weeks) per genotype in two independent experiments. Final Foxp3% after *in vitro* expansion was 74.45% ± 8.03 for *Ndfip1$^{fl/fl}$ Foxp3*-Cre versus 74.8 ± 9.80 for *Ndfip1$^{+/+}$ Foxp3*-Cre. (**e–h**) Represents $n = 6$–8 mice (male and female, 7–12 weeks) in three experiments. Final Foxp3% was 88.1 ± 2.08 for *Ndfip1$^{fl/fl}$ Foxp3*-Cre versus 86.43 ± 1.94 for *Ndfip1$^{+/+}$ Foxp3*-Cre. P values were calculated by one-way ANOVA. *$P < 0.05$, **$P < 0.01$, ***$P < 0.001$, ****$P < 0.0001$.

activity, we examined other proteins in the mTORC1 pathway, including the amino acid transporter, CD98, and phosphorylated S6 (pS6). CD98 and pS6 (Fig. 8f,g) were both increased in *Ndfip1$^{fl/fl}$ Foxp3*-Cre $T_{reg}$ cells. Coculture confirmed that these changes are not directly linked to IL-4 signalling (Fig. 8h).

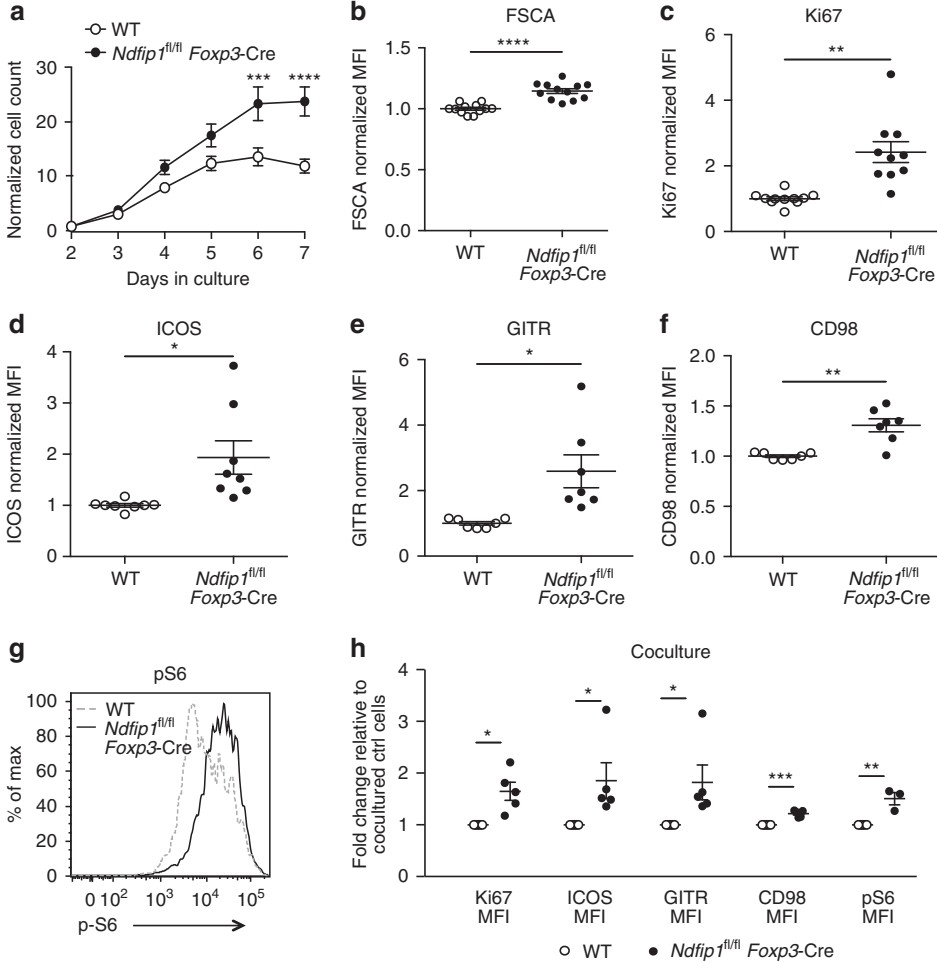

**Figure 8 | Loss of Ndfip1 leads to elevated mTORC1 signalling and metabolic fitness in T_reg cells. (a–h)** T_reg cells were sorted from congenically distinct CD45.1 WT or CD45.2 Ndfip1^fl/fl Foxp3-Cre mice. T_reg cells were expanded in vitro (**a–g**) individually or (**h**) in mixed cocultures. (**a**) Cell numbers in the cultures were analysed daily for 7 days. (**b–f**) After 7 days, cells were analysed by flow cytometry for cell size (forward scatter (FSC)), (**b**) Ki67 expression, (**c**) or their surface levels of: (**d**) ICOS, (**e**) CD98, and (**f**) GITR. (**g**) Cells were also restimulated at the end of the culture period and analysed for expression of pS6. (**h**) Cocultured WT and Ndfip1^fl/fl Foxp3-Cre T_reg cells were similarly examined for the expression of Ki67, ICOS, GITR, CD98 and pS6. Each dot represents an individual mouse. All experiments were performed on at least two independent occasions using age-matched male or female, aged 7–12 weeks, mice of each genotype. P values were calculated by multiple t-tests. *P < 0.05, **P < 0.01, ***P < 0.001, ****P < 0.0001. Error bars indicate mean ± s.e.m.

Increased responsiveness to IL-2 could explain the fitness of Ndfip1-deficient T_reg cells in this IL-2 expansion system. To test this, we sorted Ndfip1-sufficient and -deficient T_reg cells and rested them overnight without cytokine or TCR stimulation. Upon brief IL-2 stimulation, we detected STAT5 phosphorylation (pSTAT5) by flow cytometry. pSTAT5 levels (mean fluorescence intensity) were similar between WT and Ndfip1-deficient T_reg cells before and after IL-2 stimulation (Supplementary Fig. 8). This is consistent with our recent report showing that both Ndfip1 and its homologue, Ndfip2, are required in T_conv cells to drive the degradation of Jak1 and decrease STAT5 phosphorylation in response to IL-2 (ref. 40). Therefore, the increased metabolic activity of Ndfip1^fl/fl Foxp3-Cre^+ T_reg cells is cell intrinsic and not just because of enhanced responsiveness to IL-2.

## Discussion

Therapies using adoptively transferred T_reg cells or that target T_reg cell function are now being developed, making it essential to understand mechanisms that maintain T_reg cell identity and how these pathways integrate with TCR activation and cellular

metabolic processes. Our data identify Ndfip1, an activator of E3 ubiquitin ligases, as a new molecular target impacting T_reg stability and function. We demonstrate that Ndfip1-deficient T_reg cells are highly proliferative and concomitantly gain the capacity to produce the proinflammatory cytokine IL-4. Importantly, Ndfip1 deficiency can lead to the expansion of T_reg cells and loss of Foxp3 in vivo.

Both mTORC1 and mTORC2 play important functions in preventing T_reg cells from adopting a glycolytic metabolic profile. mTORC1-mediated glycolytic activity in T_reg cells is modulated by the serine–threonine phosphatase, PP2A, and by the autophagy proteins, Atg5 and Atg7 (refs 7,8). The lipid phosphatase, PTEN, inhibits mTORC2-mediated metabolic pathways in T_reg cells to prevent T_reg cell instability and maintain suppression of Th1 and Tfh effector T-cell functions[41,42]. Ndfip1-deficient T_reg cells have increased protein levels of ICOS, CD98 and pS6, all indicators of high mTORC1 signalling. mTORC1 deletion in T_reg cells leads to defects in efficiently synthesizing lipids from glucose and to defects in homeostatic proliferation in vivo[39]. Therefore, it is not surprising that Ndfip1-deficient T_reg cells, with high mTORC1 signature,

show greater homeostatic expansion *in vivo*, show greater metabolic fitness *in vitro* and *in vivo* and are more efficient at glycolysis. This altered glycolytic metabolism fuels dysfunction *in vivo* by driving the expansion of eT$_{reg}$ cells that are prone to produce IL-4 and lose Foxp3.

We propose that T$_{reg}$ cells lacking Ndfip1 progress through a series of changes that is precipitated by increased proliferation of activated (CD44$^+$) T$_{reg}$ cells and accompanied by the acquisition of IL-4 production. These changes are likely initiated in eT$_{reg}$ cells since Ndfip1 is inducibly expressed upon T-cell activation. The elevated glycolytic metabolism in these cells drives further expansion of IL-4-producing current and former T$_{reg}$ cells. Autocrine IL-4 signalling can then lead to loss of Foxp3. Supporting this, CD44$^+$ Ndfip1-deficient T$_{reg}$ cells have increased methylation at the CNS2 region of the *Foxp3* locus, indicating susceptibility towards instability. Ndfip1-deficient T$_H$2 cells are capable of driving tissue damage and inflammation as already described by our laboratory and others[12,14,40,43].

Loss of IL-4 did not affect the activation, proliferation and *in vivo* expansion of Ndfip1-deficient T$_{reg}$ cells leading us to conclude that the metabolic fitness in Ndfip1-deficient T$_{reg}$ cells precedes or is independent of the gain in T$_{reg}$ cell-intrinsic IL-4 production. Furthermore, IL-4-overexpressing transgenic mice have been described and have increased recruitment of different immune cell subsets into the skin but do not develop spontaneous dermatitis[44] as observed in *Ndfip1*$^{fl/fl}$ *Foxp3*-Cre mice. Last, loss of the mTORC1 subunit, Raptor, in T$_{reg}$ cells does not lead to a cell-intrinsic increase in T$_{reg}$ cell IL-4 or other effector cytokines[39], suggesting that cytokine regulation and metabolic regulation by Raptor in T$_{reg}$ cells may be two independent events[39]. An attractive hypothesis that warrants further future investigation is whether two parallel (but perhaps cross-talking) cellular pathways are regulated by Ndfip1: one which dampens mTORC1 activity and glycolysis in order to limit T$_{reg}$ cell activation and proliferation, and a second that limits T$_{reg}$ cell IL-4 production. These pathways, though independent, may cooperate to account for the severity of the inflammation observed in *Ndfip1*$^{fl/fl}$ *Foxp3*-Cre mice.

Ndfip1 is a known activator of Itch[40,45,46]. Mice bearing a T$_{reg}$-specific deletion of *Itch* show some overlapping features with the *Ndfip1*$^{fl/fl}$ *Foxp3*-Cre mice described here[15]: Itch-deficient T$_{reg}$ cells have an effector T$_{reg}$ cell phenotype, an increase in proliferation and intact suppression in traditional *in vivo* and *in vitro* suppression assays. Itch-deficient T$_{reg}$ cells are unstable as defined by their increased production of IL-4 but these T$_{reg}$ cells do not have defects in their ability to maintain Foxp3 expression in a lymphoreplete animal[15]. Future work will be needed to determine whether mTORC1 signalling is elevated in Itch-deficient T$_{reg}$ cells and whether this contributes to their dysfunction. Future studies on Itch and Ndfip1 will be of particular interest as therapies that activate or inhibit these pathways are being developed. If such therapeutic strategies can be used to regulate T$_{reg}$ cell functions, their use in the clinic could have broad potential.

## Methods

**Mouse strains.** *Ndfip1*$^{fl/fl}$ mice[47] were crossed to *Foxp3-YFP-Cre* reporter mice (Jackson Laboratory, stock# 016959[17]), also referred to as *Foxp3*-Cre, and bred for only one copy of the Cre gene to generate *Ndfip1*$^{fl/fl}$ *Foxp3*-Cre male mice and *Ndfip1*$^{fl/fl}$ *Foxp3*-Cre$^{+/-}$ female mice. WT control mice were: *Ndfip1*$^{+/+}$ *Foxp3*-Cre$^+$ male, and *Ndfip1*$^{+/+}$ *Foxp3*-Cre$^{+/-}$ female mice. We generated congenic CD45.1/CD45.2 control mice by crossing *Ndfip1*$^{+/+}$ *Foxp3*-Cre$^+$ mice to CD45.1 mice (Jackson Laboratory, Stock# 002014). CD45.1 and *Rag1*$^{-/-}$ are maintained in house. *Ndfip1 IL-4* DKO mice[25,26] are maintained in house. The gender and ages of mice used are indicated in each figure legend. Mice were maintained in a barrier facility at the Children's Hospital of Philadelphia. All procedures were approved by the Institutional Care and Use Committee of the Children's Hospital of Philadelphia. Genotyping primer sequences used were:

*Ndfip1* flox alleles (*Ndfip1* floxed forward 5′-TGAGGAAACAGACACACAATG-3′, *Ndfip1* floxed reverse 5′-TGGAATGAACCTGAGGTCTCC-3′)[48] and *Foxp3*-Cre (Jackson Laboratory, stock# 016959, WT Forward 5′-CCTAGCCCCTAGTTCCA ACC-3′, WT Reverse 5′-AAGGTTCCAGTGCTGTTGCT-3′, Mutant Forward: 5′-AGGATGTGAGGGACTACCTCCTGTA-3′, Mutant Reverse 5′-TCCTTCACT CTGATTCTGGCAATTT-3′)[17]. Sequences for *Il4* genotyping (Jackson Laboratory, stock# 002253) are: *Il4* common 5′-GTGAGCAGATGACATGGGGC-3′; *Il4* WT 5′-CTTCAAGCATGGAGTTTTCCC-3′ and *Il4* mutant 5′-GCCGATTGTCTGTT GTGCCCAG-3′.

**Flow cytometry.** The following flow cytometric antibodies were purchased from Biolegend: CD4 (GK1.5), CD8α (53-6.7), CD44 (IM7), IL-17A (TC11-18H10.1), CD3ε (17A2), CD45.2 (104), CD45.1 (A20) GITR (YGITR765), ICOS (C398.4A), and PD-1 (RMP1-30), all used at 1:200 dilution. αYFP/GFP (A-21311) was purchased from Life Technologies and used at 1:400 dilution. Antibodies against IFNγ(XMG1.2) used at 1:300 and Ki67 (B56) used at 1:200 dilution were purchased from BD Biosciences. Biotinylated αphospho-S6 was purchased from Cell Signalling Technologies (D57.2.2E) and used at 1:150 dilution. The remaining antibodies for flow cytometry were purchased from eBioscience: CD25 (PC61.5), CD62L (MEL-14), and IL-10 (JES5-16E3) all used at 1:200 dilution; IL-4 (11B11) used at 1:100 dilution; and Foxp3-biotin (FJK-16s). Biotinylated αFoxp3 and αpS6 were used at 1:150 dilution and detected with fluorophore-conjugated streptavidin at 1:750 dilution (S32354 Invitrogen).

**Tissue processing and flow cytometry.** Serum was obtained by cardiac puncture immediately following CO$_2$ killing. Spleens and lymph nodes were harvested and mashed through 70 μM nylon filters using cold Hank's Balanced Salt Solution (HBSS). Spleens were red blood cell (RBC) lysed using ammonium-chloride-potassium lysis buffer, washed, filtered again via a 70 μM filter and resuspended in cold HBSS for sorting or resuspended in complete DMEM (cDMEM) for either phorbol myristate acetate (PMA)/Ionomycin stimulation or direct flow cytometry. Lungs were flushed with cold phosphate-buffered saline (PBS) immediately after killing, minced in cDMEM medium containing 0.9 mg ml$^{-1}$ of collagenase A (Sigma), 0.8 mg ml$^{-1}$ of collagenase 1A (Sigma) and 20 μg ml$^{-1}$ of DNase I (Sigma) and rotated for 1 h at room temperature. FCS was added at 15% v/v and the mixture was filtered through a 100 μM and then 40 μM filter before centrifugation. Lung pellets were RBC lysed using in-house-made ammonium-chloride-potassium lysis buffer, washed, resuspended in complete DMEM media and filtered through a 70 μm filter. The cells were then stimulated for 4 h in 5 ml round bottom polystyrene test tubes with PMA (30 ng ml$^{-1}$, Calbiochem), ionomycin (1 μM, Abcam) and Brefeldin A (1 μg ml$^{-1}$, Sigma) for intracellular cytokine staining or directly stained for flow cytometry. For flow cytometry, cells were washed in serum-free HBSS or PBS, stained with live/dead fixable blue dead cell stain (L-23105; Invitrogen, 1:60 dilution), Fc blocked (αCD16/32, 2.4G2; BD Biosciences, 1:25 dilution) and stained with the appropriate antibody mixtures. After staining 25 min at 4 °C, samples were washed in FACs buffer (1xPBS, 2.5% v/v FCS and 0.1%w/v Sodium azide) and then fixed overnight using the Foxp3 Fix/Perm Kit (00-5523-00, eBioscience). After washing with the provided Foxp3 kit perm buffer, intracellular staining was carried out for 1 h at 4 °C. Cells were washed again with perm buffer and resuspended for flow cytometry in FACs buffer. In Fig. 8, expanded T$_{reg}$ cells, cultured as described below, were restimulated with mouse T-cell activator beads (Gibco, 11456D) at a 1:1 cell:bead ratio in complete DMEM medium for 4 h. Samples were immediately fixed at a final concentration of 2% PFA, permeabilized in 90% methanol and then stained for pS6 and surface markers together in Fc block. The same PFA fixation protocol was used for the pSTAT5 experiment. Samples were acquired on an LSRFortessa (BD Biosciences) and analysed using FlowJo version 9.8 (Flowjo LLC). Events analysed were singlets, (FSC-A × FSC-H, SSC-W × SSC-H), live (viability dye negative) and within the lymphocyte gate (FSC-A × SSC-A) (Supplementary Fig. 9).

**Enzyme-linked immunosorbent assay (ELISA).** Serum was isolated from 8- to 16-week-old male Ndfip1-sufficient and -deficient Foxp3-Cre$^+$ mice. The concentrations of Ig subclasses in mouse sera were determined using isotype-specific antibodies with a sandwich ELISA protocol. Monoclonal anti-mouse IgA-HRP, IgM-HRP, IgG1-HRP, IgG2C-HRP and IgE-HRP were purchased from SouthernBiotech. These were used in twofold serial dilutions starting at 10 ng ml$^{-1}$. Flat-bottom, 96-well plate (Nunc) was coated with capture antibody overnight at 1:250 dilution. Plates were blocked with 10% FCS in PBS buffer for 2 h and incubated with sample serum (1:20,000 for IgA, IgM and IgE and 1:40,000 for IgG1 and IgG2C) for 2 h at room temperature and detected with HRP-conjugated Ig subclass antibody (SouthernBiotech, 1:10,000 dilution) for 1 h at room temperature. Plates were developed with TMB substrate solution (eBioscience) and read at 450 nM using a Synergy HT Microplate Reader (BioTek).

**T-cell isolation.** YFP$^+$ CD4$^+$ T$_{reg}$ cells were isolated from the spleen and lymph node, enriched for CD4$^+$ T cells by negative selection using 5 μl per mouse of rat αmouse CD8α (2.43; Biolegend,) and 5 μl per mouse of rat αmouse I-A/I-E (M5/114.15.2; Biolegend) in 5–7.5 ml of complete DMEM medium and incubated for 30 min with end-over-end rotation at 4 °C. Cells were then washed and

incubated with 1.5 ml of Biomag goat αrat IgG magnetic beads (Qiagen) for 10 min with end-over-end rotation at room temperature; unbound cells were collected using a Dynal magnet (Invitrogen). These enriched and untouched $CD4^+$ T cells were then stained with antibodies against CD4, CD44, CD25 and CD62L all at 1:200 dilution. Cells were filtered through 35 μM filter cap polystyrene FACS tubes (BD Biosciences) and sorted under high speed on a MoFlow Astrios (Beckman Coulter) or a FACS Aria (BD Biosciences). Total $T_{reg}$ cells were sorted as $CD4^+$ $YFP^+$, $cT_{reg}$ cells as $CD4^+YFP^+$ $CD62L^{high}CD44^-$ and $eT_{reg}$ cells as $CD4^+YFP^+$ $CD62L^{low}CD44^+$. Naïve cells were identified as $CD4^+CD62L^{high}CD44^-CD25^-$.

**Bisulfite sequencing.** Sodium bisulfite conversion, followed by PCR amplification and sequencing of individual clones, was used to determine the extent of CpG DNA methylation at the CNS2 and promoter regions of the Foxp3 gene. First, $CD4^+YFP^+$ $CD62L^{low}CD44^+eT_{reg}$ cells were sorted from $YFP^+$ WT and $Ndfip1^{fl/fl}$ Foxp3-Cre$^+$ mice. Naïve $CD4^+$ T cells were sorted from WT mice. Next, DNA was extracted using the Qiagen DNeasy Blood and Tissue Kit. Approximately 1 μg of DNA purified from $CD4^+$ $T_{reg}$ cells was bisulfite converted via the following procedure[49]: the DNA was precipitated in ethanol, washed with 70% ethanol, and resuspended in 20 μl of Tris-EDTA buffer. This DNA was then denatured at 37 °C for 15 min using 0.3 M NaOH and then incubated at 55 °C for 16 h in a mixture containing 5.36 M Urea, 1.72 M sodium metabisulfite and 0.5 mM hydroquinone. The DNA was then desalted using the Wizard DNA clean-up system (Promega, A7170), desulfonated with 0.3 M NaOH, neutralized with 3 M ammonium acetate and precipitated with 100% ethanol. The CNS2 region of the Foxp3 gene was PCR amplified using the following primers: Foxp3 Intron Forward1 5′-GGGTTTTGGGATATTAATATATATAGTAAG-3′, Foxp3 Intron Reverse1 5′-CCACTATATTAACTTAACCCATATAACTAA-3′, Foxp3 Intron Forward2 5′-TTGAGTTTTTGTTATTATAGTATTTGAAGAT-3′, and Foxp3 Intron Reverse2 5′-ACTAAAAACCTAAAAAACTAAACTAACCAA-3′. The promoter region was amplified with the following primers: Foxp3 Promoter Forward1 5′-GTTTGGAGTAGAAGGAAGTTTTTGGAGAT-3′, Foxp3 Promoter Reverse1 5′-TATCTAAAAACCAACTACTCCACCTATCTA-3′, Foxp3 Promoter Forward2 5′-GGTTGTTTTTTATTTATATGGTAGGT-3′, and Foxp3 Promoter Reverse2 5′-CCAAAATCCTTACCTAAAATAACTA-3′. Following nested PCR, bands of appropriate sizes were gel purified and cloned into pGEM-T Easy vector (Promega, A1360). Single colonies were miniprepped with the QIAprep Spin Miniprep DNA Purification Kit (Qiagen, 27104) and sequenced with SP6 primer. Only sequences that were derived from fully converted alleles were used for methylation analysis.

***In vitro* cell culture.** For the in vitro $T_{reg}$ cell suppression assay, sorted naïve $T_{conv}$ $CD45.1^+$ $CD4^+CD62L^{high}CD44^-CD25^-$ cells were stained with 2.5 μM carboxyfluorescein succinimidyl ester (CFSE) by incubating 400 μl of cells (10-20 million cells per ml) in PBS with 400 μl of 5 mM CFSE in PBS solution for 3 min with continuous shaking. Next, 800 μl of FCS was added to the reaction mixture for 30 s at room temperature, followed by adding 10 ml of complete DMEM medium and spinning down in a centrifuge at 1300 r.p.m. for 5 min at 4–6 °C. The CFSE-labelled naïve $T_{conv}$ cells were then resuspended to achieve $1.5 \times 10^4$ cells per 50 μl, and 50 μl of these cells were added to the appropriate wells of a 96-well plate. Sorted $YFP^+$ $CD4^+CD45.2^+$ $T_{reg}$ cells were resuspended at $1.5 \times 10^4$ cells per 50 μl. A total of 50 μl of twofold serial dilutions of these $T_{reg}$ cells were added to the $T_{conv}$ cells in each well of the 96-well plate to achieve a final $T_{reg}:T_{conv}$ ratio per well ranging from 0:1 to 1:32 (ref. 50). Next, irradiated bulk $CD45.2^+$ splenocytes (2,500 rads, X-ray irradiator) were resuspended at $1.5 \times 10^5$ cells per 50 and 50 μl was added. Last, 50 μl of a $4\times$ solution containing αCD3 (145-2C11; Biolegend) and rhIL-2 were added to achieve a final concentration of 1 μg ml$^{-1}$ soluble αCD3 and 50 U ml$^{-1}$ IL-2 per well. The cells were cultured for 4 days for the suppression assay. For expanded $T_{reg}$ cultures, sorted $YFP^+$ $CD4^+$ total $T_{reg}$ cells or $CD4^+YFP^+$ $CD62L^{high}CD44^-$ $cT_{reg}$ cells were resuspended at $1 \times 10^6$ cells per ml in media containing 2,000 U ml$^{-1}$ IL-2 and stimulated at a 1:3 (cell:bead) ratio with mouse αCD3/αCD28 T-cell activator Dynabeads (Gibco, 11456D). After 48 h, cultures were counted and split daily with IL-2 media. Cells were assayed by flow cytometry after 6 or 7 days in cultures. Due to the 48-h activation, results from experiments where we started with sorted $YFP^+$ $cT_{reg}$ cells were virtually identical to those in which we started with sorted $YFP^+$ total $T_{reg}$ cells due to in vitro conversion of c $T_{reg}$ to e $T_{reg}$ cells. We therefore present all of that data pooled together. For analysis of RNA, expanded $T_{reg}$ cells were restimulated on day 7 of culture at a 1:1 ratio with αCD3/αCD28 beads for the indicated periods. Supernatant was saved for ELISA and cells were harvested for quantitative PCR (qPCR). For cocultures, sorted WT and Ndfip-1 deficient $T_{reg}$ cells were mixed in a 1:1 ratio before being plated. For the in vitro stability assay using IL-2, IL-4 and αIL-2, WT or Ndfip1-deficient $YFP^+$ $T_{reg}$ cells were sorted from matched male or female donors. The $T_{reg}$ cells were cultured for 96 h on plate-bound 5 μg ml$^{-1}$ αCD3/αCD28 and either 500 U ml$^{-1}$ IL-2 or 20 ng ml$^{-1}$ IL-4 with 20 ng ml$^{-1}$ α-IL-2. All cultured cells, unless otherwise noted, were cultured at 10% $CO_2$ in complete T-cell medium, that is, DMEM (Mediatech) supplemented with 10% FCS (Atlanta Biologicals, premium FCS), 1% pen/strep (Invitrogen), 1% Glutamax (Invitrogen), 1% Minimum Essential Medium–non-essential amino acids (Invitrogen), 20 mM HEPES (Invitrogen), 1 mM Sodium Pyruvate and 0.12 mM betamercaptoethanol (Sigma) supplemented with

recombinant human 50 U ml$^{-1}$ IL-2 (originally obtained through the AIDS Research and Reference Reagent Program, Division of AIDS, National Institute of Allergy and Infectious Diseases, National Institutes of Health).

**Metabolic function assays.** $YFP^+$ $CD44^-CD62L^+$ $T_{reg}$ cells ($cT_{reg}$ cells ) or $YFP^+$ $CD4^+$ $T_{reg}$ cells (total $T_{reg}$ cells ) were sorted from 7- to 12-week-old, age-matched male or female, congenically marked CD45.1 WT or CD45.2 $Ndfip1^{fl/fl}$ Foxp3-Cre$^+$ (cKO) mice. After sorting, the WT or cKO $T_{reg}$ cells were expanded at a 1:3 (cell:bead) ratio with mouse αCD3/αCD28 beads for 7 days. On day 7, cells were harvested and resuspended at $5 \times 10^6$ cells per ml in appropriate Seahorse media. In all, 40 μl of cells were added to 140 μl of appropriate seahorse media or 140 μl of Seahorse media containing fresh mouse αCD3/αCD28 beads at a 1:1 (cell:bead) ratio. Cells were plated in XF96 cell culture microplates that were precoated with 25 μl of Cell tak reagent overnight according to the manufacturer's instructions (Corning, 354240).

For examining glycolysis, the glyco-stress test kit (Seahorse Bioscience) was used with additions of the following reagents to cells in glucose-free Seahorse media: 10 mM glucose, 2 μM oligomycin, and 50 mM 2-DG. To determine the rate of glycolysis, $T_{reg}$ cells were incubated in glucose-free medium and the increased extracellular acidification rate response upon stimulation of glycolysis by glucose and oligomycin was determined. Oligomycin inhibits mitochondrial ATP synthase (complex V) forcing glycolysis to compensate for the lack of ATP production in oxidative phosphorylation. Finally, 2-dexoglucose was added to inhibit glycolysis. For examining mitochondrial function, a mito-stress kit was used with additions of the following reagents to cells in appropriate Seahorse media: 2 μM Oligomycin, 1.5 μM fluoro-carbonyl cyanide phenylhydrazone (an uncoupler of respiration and oxidative ATP synthesis), and premixed 100 nM rotenone + 1 μM antimycin A solution (Electron Transport Chain complex I and III inhibitors). Plates were run on an XF96 Seahorse assay instrument. Settings used were: calibration, equilibration, and baseline readings (loop three times). Before the baseline reading, and after each injection from a port, the following procedure was followed: mix, wait 3 min, and measure for 2 min with 3 min end loop[51]. Each reagent was added according to the volumes recommended by the Seahorse assay manufacturer.

**Quantitative PCR.** Samples for qPCR were lysed in Trizol (Ambion). RNA was extracted using chloroform and qPCR was performed as follows: 10 ng of cDNA was added to TaqMan Gene Expression Master Mix and TaqMan Gene Expression primer/probe mix specific for Ndfip1 (Applied Biosystems), according to the manufacturer's protocol for a final reaction volume of 20 μl. qPCR was performed using an Applied Biosystems 7500 Real-Time PCR system. Each sample was assayed in triplicate along with the endogenous control (actin). Actb primer/probe (4352933E) was obtained from Applied Biosystems. Ndfip1 custom primers: Ndfip1_F 5′-GCTCCTCCACCATACAGCAGC-3′; Ndfip1_R 5′-CGATGGGGGCT TTGGAAATCCAG-3′, and Ndfip1 Taqman MGB probe: 5′-TTTGGAAATCCAG ATTCATCTTTG-3′ were obtained from Applied Biosystems[40]. Relative mRNA expression of each gene of interest was calculated as $2^{dCt}$, where dCt represents threshold cycle (Ct) of Actin beta minus Ct of gene of interest.

**Bone marrow chimeras.** Bone marrow from male $Ndfip1^{fl/fl}$ Foxp3-Cre$^+$ (CD45.2$^+$) or WT mice: $Ndfip1^{+/+}$ Foxp3-Cre$^+$ (CD45.1/CD45.2) or CD45.1$^+$ mice was obtained by flushing femur and tibia with cold HBSS, RBC lysing and passing through a 70 μM filter. Cells were processed into a single-cell suspension, T cell depleted, resuspended in freezing media (90% FCS, 10% DMSO) and kept at −80 °C until used. Thawed cells were washed, counted, resuspended in sterile PBS, mixed 1:1 CD45.1:CD45.2 and injected intravenously into sublethally irradiated (400 rads, X-Ray irradiator) 6-week-old male Rag1$^{-/-}$ recipients at $10 \times 10^6$ cells per mouse. Chimeras were kept on 5.3 ml of Trimethoprim/Sulfamethoxazole (200 mg sulfamethoxazole + 40 mg trimethoprim per 5 ml) per 400 ml drinking water for the first 2 weeks after bone marrow transfer. Chimeras were analysed 8 weeks after transfer.

**Histology.** Skin (ear) and oesophagus were dissected and fixed in 10% neutral buffered formalin for at least 24 h. Lung were obtained after manual transcardial perfusion with 10 ml syringes containing PBS and fixed in 10% neutral buffered formalin. All organs were then embedded in paraffin, sectioned to 5 μm thickness and stained with Haematoxylin and Eosin. Images were obtained using a Leica DM4000B upright scope paired with a Spot RT/SE Slider camera (Children's Hospital of Philadelphia Pathology Core).

**T-cell transfer colitis.** Naïve $CD4^+$ T cells from WT mice (CD45.1$^+$ or CD45.1/CD45.2) and $CD4^+YFP^+$ $T_{reg}$ cells from CD45.2$^+$ $Ndfip1^{fl/fl}$ Foxp3-Cre and CD45.2$^+$ $Ndfip1^{+/+}$ Foxp3-Cre male mice were isolated from the spleens and lymph nodes. Cells were resuspended in sterile PBS and congenic naïve T cells were mixed at a 5:1 ratio with $YFP^+$ $T_{reg}$ cells. A total of $0.36 \times 10^6$ cells were injected intraperitoneally into 6–8-week-old Rag1$^{-/-}$ mice. Mice were weighed weekly. Recipients of both genotypes were cohoused. At killing, mice were weighed, spleens were weighed and the spleens and lung were processed for flow cytometry as described above. A fraction of the total lung/spleen single-cell suspensions were

stained with antibodies against CD4, CD45.1 and CD45.2 and sorted using a Moflow Astrios as described above for analysis of Ndfip1 mRNA in cells from $Ndfip1^{fl/fl}$ $Foxp3$-Cre and $Ndfip1^{+/+}$ $Foxp3$-Cre donor $T_{reg}$ cells. Sorted cells were CD4$^+$CD45.2$^+$CD45.1$^-$YFP$^+$ (current $T_{reg}$ cells ) or CD4$^+$CD45.2$^+$ CD45.1$^-$YFP$^-$ (former $T_{reg}$ cells) or CD4$^+$CD45.2$^-$CD45.1$^+$ YFP- ($T_{conv}$ cells). Cells were stimulated for 5 h with plate-bound 5 µg ml$^{-1}$ αCD3/CD28 at $3 \times 10^5$ cells per ml and harvested in Trizol for qPCR. The remaining fractions of total lung/spleen single-cell suspensions were either stained directly for flow cytometry or stimulated for 4 h with PMA/ionomycin in the presence of Brefeldin A to analyse cytokine production by flow cytometry.

**Preparation of lysates for whole-cell proteome analysis.** Ndfip1-sufficient and -deficient c$T_{reg}$ cells and e$T_{reg}$ cells were isolated and lysed using a lysis buffer containing 100 mM Tris-HCl, pH 8.0, 0.15 M NaCl, 5 mM EDTA, 1% NP-40, 0.5% Triton-X 100, protease inhibitor cocktail (Roche, 11697498001), 5 mM of 1,10-phenyanthroline (o-PA), 5 mM N-ethylmaleimide and 0.1 mM PR-619. In all, 10 µl of lysis buffer was used for every $10 \times 10^6$ cells. Protein was quantified by BCA (Pierce, 23227). Approximately 15 µg of lysate was mixed 1:1 with $4 \times$ Laemmli sample buffer for whole proteome analysis. Samples were boiled and run $\sim 2$ cm past the stacking gel in 10% Criterion precast Tris-HCl gels (Biorad). Gels were fixed overnight and stained briefly with Coomassie blue. Each lane of the Coomassie-stained gel was divided into 8–10 $2 \times 9$ mm$^2$ 'pixels,' each cut into 1 mm$^3$ cubes[52]. They were destained with 50% Methanol/1.25% Acetic Acid, reduced with 5 mM dithiothreitol (Thermo) and alkylated with 40 mM iodoacetamide (Sigma). Gel pieces were then washed with 20 mM ammonium bicarbonate (Sigma) and dehydrated with acetonitrile (Fisher). Trypsin (5 ng µl$^{-1}$ in 20 mM ammonium bicarbonate, Promega) was added to the gel pieces and proteolysis was allowed to proceed overnight at 37 °C. Peptides were extracted with 0.3% triflouroacetic acid (J.T. Baker), followed by 50% acetonitrile. Extracts were combined and the volume was reduced by vacuum centrifugation.

**Mass spectrometric analysis.** Tryptic digests were analysed by LC-MS/MS on a hybrid LTQ Orbitrap Elite mass spectrometer coupled with a Dionex Ultimate 3000 (Thermofisher Scientific San Jose, CA). Peptides were separated by reverse phase (RP) HPLC on a nanocapillary column, 75 µm id $\times$ 30 cm Reprosil-pur 1.9 µM (Dr Maisch, Germany). Mobile phase A consisted of 0.1% formic acid (Thermo) and mobile phase B of 0.1% formic in acetonitrile. Peptides were eluted into the mass spectrometer at 300 nl min$^{-1}$ with each RP-LC run comprising a 90 min gradient from 3% to 45% B in 90 min. The mass spectrometer was set to repetitively scan m/z from 300 to 1,800 ($R = 240,000$ for LTQ-Orbitrap Elite) followed by data-dependent MS/MS scans on the 20 most abundant ions, with a minimum signal of 1,500, dynamic exclusion with a repeat count of 1, repeat duration of 30 s, exclusion size of 5,000 and duration of 60 s, isolation width of 2.0, normalized collision energy of 33 and waveform injection and dynamic exclusion enabled. Fourier transform MS full-scan AGC target value was 1e6, while MSn AGC was 1e4. Fourier transform MS full-scan maximum fill time was 10 ms, while ion trap MSn fill time was 100 ms; microscans were set at one. FT preview mode; charge state screening and monoisotopic precursor selection were all enabled with rejection of unassigned and 1+ charge states.

**Proteomic data analysis.** Proteomic data were analysed using Maxquant version 1.5.0.30 searching against the Uniprot complete mouse reference proteome, including isoforms, (updated 19 September 2013) and common laboratory contaminants. A minimum peptide length of six amino acids and a peptide and protein false discovery of 1% was required. The three biological replicates for Ndfip1-sufficient and -deficient e$T_{reg}$ cells and c$T_{reg}$ cells were analysed together, with match between runs and requantify turned on. Label-free quantification via intensity-based absolute quantification was calculated in MaxQuant. Quantification data were normalized by the mean quantification value of identified proteins in each replicate and log2 transformed. Comparisons of protein hits from WT or Ndfip1-deficient e$T_{reg}$ or c$T_{reg}$ cells were calculated by log2 fold changes and evaluated for significance using a one-sample t-test. Area-proportional Venn diagrams were generated with eulerAPE version 3.0 (http://www.eulerdiagrams.org/eulerAPE/)[53]. The GO analysis and corresponding network diagram in Fig. 6e was generated with the BiNGO application[54] in Cytoscape[55]. The differentially regulated proteins identified in WT vs cKO e$T_{reg}$ comparisons were compared against the *Mus musculus* GO Biological Process annotation using a hypergeometric significance test with Benjamini–Hochberg false-discovery rate correction. A threshold of adjusted P value of <0.01 was required for enrichment.

**Statistical analysis.** Data were graphed and analysed for statistical significance in Prism version 6 or version 7 (Graphpad Software, Inc) or Excel (Microsoft). The heatmap in Fig. 5 was generated in prism 7. The following statistical tests were used as appropriate and as noted in the figure legends: t-test, one-way analysis of variance (ANOVA), two-way ANOVA, and repeated-measures ANOVA. All data are shown as average ± s.e.m., with a cutoff of $P < 0.05$ for statistical significance: *$P < 0.05$, **$P < 0.01$, ***$P < 0.001$, ****$P < 0.0001$.

**Data availability.** All data generated or analysed during this study are included in this published article (and its Supplementary Information files). The mass spectrometric proteomics data have been deposited to the ProteomeXchange Consortium via the PRIDE[56] partner repository with the data set identifier PXD006251.

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

## Acknowledgements

We thank Steve Seeholzer, Hossein Fazelinia, Hua Ding and Lynn Spruce of the Children's Hospital of Philadelphia Research protein core. We thank Stephanie Sprout for mouse colony management. This work was supported by the National Institutes of Health Grants R01AI093566 and R01AI114515 to P.M.O.; T32CA009140-39 to A.A.K.L.; the American Asthma Foundation Grant 13-0020 to P.M.O.; and by the University of Pennsylvania Diabetes Center Islet Cell Biology Core P30-DK19525 to N.M.D.

## Author contributions

A.A.K.L., G.D., C.E.O., S.T., R.M.T. and N.M.D. performed experiments, analysed data and/or assembled figures. N.M.D. carried out the metabolic function assays. E.K.M. assisted with *in vivo* injections and discussions. J.M.D. assisted with proteomic bioinformatics analysis. A.A.K.L. wrote the manuscript. P.M.O. conceived the project, provided guidance and edited the manuscript. All authors read/edited the manuscript.

## Additional information

**Competing interests:** The authors declare no competing financial interests.

