## [Peer Review File · Nature Communications]

Reviewers' comments:

Reviewer #1 (Remarks to the Author):

Deng et al study the functions of Ndfip1, an activator of Nedd4-family ubiquitin ligases, including Itch. While the Oliver lab had previously shown that Ndfip1 was important for the differentiation of Treg cells induced by TGF β , whether Ndfip1 is important for the functions or homeostasis of differentiated Treg cells had not been examined. Using a Foxp3-Cre deletion system, Deng et al show that Treg-specific expression of Ndfip1 is important to avoid the expansion of the subset of Treg cells with effector functions (eTreg cells) and to promote stable Foxp3 expression. In addition, Ndfip1 represses IL-4 expression by Treg cells, as it does in conventional T cells, but IL-4 expression is not needed for its effect on cTreg amplification. Using metabolism (seahorse) analyses, the authors document increased glycolysis and glycolytic capacity in Ndfip1-deficient Treg cells, consistent with increased in vitro proliferation and acquisition of effector markers.

That manuscript is well presented and the data of high quality. It reports a comprehensive study of Ndfip1 functions in Treg cells, and provides insight into its impact on cell metabolism. Given the importance of these cells in immune responses, the data will be of interest to a large number of immunologists. A few suggestions below are intended to improve the manuscript.

Main comments

1-It is not clear whether Ndfip1 only serves to control the expansion of eTreg cells or if it also stabilizes cTreg cells or prevents their inappropriate conversion into eTreg cells. This could be addressed by in vitro analyses (as in Fig. 8) or adoptive transfer experiments (as in Fig. 5) performed from sorted cTreg cells.

2-The impact of Ndfip1 on metabolism could be direct or indirect, and the interpretation of the seahorse data is a bit complicated by the experimental setup (as cells need to be amplified in culture). While there is no need to expand these metabolism analyses in the current report, the author's recent demonstration that Ndfip molecules promote the degradation of Jak kinases raises the obvious possibility that increased metabolism in Ndfip1-deficient cells results from increased cytokine signaling. Is there any evidence for increased Jak expression or Stat phosphorylation in the mutant Treg cells?

Minor comments:

-Do the mutant mice die prematurely?

-In figures legends or methods, it would be useful to state whether animals used in experiments are male or female. This can otherwise be confusing (e.g. Fig. 4, bone marrow chimeras where one assumes donors are male).

-Lines 168 and following, the text does not really describe the data shown in figures 3h-l, and is confusing when compared to Figs. S2c-f. The text could be clarified to better fit with the actual figures.

-Line 261: the subtitle does not fit with the actual data (which shows that IL-4 is not necessary).

-Line 294, Lag3 is not the official gene symbol for Galectin3.

-In Fig. 7, what is the percentage of cells that remain Foxp3+ at the end of the culture?

Reviewer #2 (Remarks to the Author):

In their manuscript, "Ndfip1 limits mTORC1 activity in regulatory T cells to prevent loss of stability and autoinflammatory disease" Deng et al. investigated the role of the Ndfip1, a co-activator of Nedd4-family, in Treg cells by conditionally deleting Ndfip1 within Foxp3+ Treg cells (using Foxp3-Cre). The loss of Ndfip1 expression within Treg cells disrupts immune homeostasis and leads to inflammation of the esophagus, skin, and lungs. This phenotype is linked to elevated IL-4 production by Treg cells, which appears to destabilize FoxP3 expression. Mechanistically, the authors demonstrated that mTORC1-related metabolic reprogramming, especially glycolysis, accounts for Treg cell instability observed in the absence of Ndfip1. In general, this study nicely detailed how Ndfip1 controls Treg cell homeostasis and function and provided new mechanistic understanding for how Treg cells can lose stability and acquire effector-like features that contribute to barrier surface pathology. Below, the authors may find comments on how to better support the overall conclusions of their manuscript:

1. As the authors state, many of the phenotypes described in the manuscript might be a secondary consequence of inflammation. To address this issue, the authors analyzed female mice that expressed a heterozygous Foxp3-Cre allele (denoted as Foxp3-Cre^{+/-} in the manuscript), but these mice appear to have inflammation similar to the male Ndfip1 conditional knockout mice (Supplemental Figure S2). Moreover, the authors state that mixed bone marrow chimeras generated with Foxp3-Cre x Ndfip1^{fl/fl} bone marrow develop signs of dermatitis by 8 weeks post bone marrow transfer. Given these data, it is difficult to rule out a role for inflammation in the phenotypic changes of Ndfip1-deficient Treg cells. To better support their conclusion that the changes are mostly attributed to loss of Ndfip1, the authors could also analyze the female Foxp3-Cre^{+/-} mice prior to the onset of inflammation (e.g. 7-10 day old mice) for their cTreg/eTreg phenotypes, ICOS, GITR, PD-1, Foxp3, and CD25 expression, as well as IL-4 production. This analysis would also allow the authors to analyze if Treg cells are lost in the periphery before inflammation, which would be predicted based on their data linking Ndfip1 to the maintenance of Treg cell stability.

2. It is quite interesting that the presence of wild-type Treg cells does not control the inflammation driven by Treg-specific deletion of Ndfip1. A similar phenomenon was also recently reported when Pten was conditionally deleted in Treg cells, where mixed bone marrow chimeras containing both wild-type and Pten-deficient Treg cells display evidence of inflammation (Shrestha et al. 2015 Nat Immunol). The conclusions from that study were that heterozygous loss of Pten is sufficient to promote Treg cell instability that ultimately led to autoimmunity. Is there any evidence the Foxp3-Cre x Ndfip1^{fl/+} show signs of inflammation or Treg cell dysfunction?

3. Because Ndfip1 deficiency might attenuate Treg cell survival or proliferation in the adoptive transfer model, the authors should validate that Treg cell stability is affected by an alternative method. For instance, their conditional knockout mice could be bred with a lineage tracing model (e.g. Rosa-linked LSL-reporter mice) and/or in vitro stability assays could be employed. These systems would allow the authors to better dissect out the roles of Treg cell proliferation, survival, and changes in Foxp3 expression to the apparent reductions of Treg cells after adoptive transfer.

4. Ndfip1-deficient Treg cells acquire the ability to produce effector cytokines, with IL-4 production being of particular interest in this study. The co-culture experiments in Figure 8h demonstrated that changes in mTORC1-related signaling appear to be independent of IL-4 signaling. However, the authors did not fully address if IL-4 could impact Treg cell instability independently of mTORC1 function. For instance, this laboratory clearly demonstrated a role for Ndfip1 in limiting IL-4 production that can suppress Foxp3 induction under iTreg cell conditions (Beal et al. 2012 Nature Immunology). Therefore, it is possible that excessive IL-4 signaling by Treg cells could contribute to the instability of Ndfip1-deficient Treg cells. To test this, the authors could perform in vitro assays where they could determine if excess IL-4 exacerbates and blocking IL-4 reduces Treg cell instability in the presence and absence of Ndfip1. Such data would be useful in further distinguishing their results from other studies, where elevated mTORC1 signaling is linked to excessive IL-17 and IFN-gamma production,

but not IL-4 production (Park et al. 2013 J Clin Invest).

5. The authors nicely demonstrate that mTORC1-related signaling and glycolysis is elevated in the absence of Ndfip1. The conclusion that mTORC1-mediated glycolysis destabilizes Treg cells in the context of Ndfip1 deficiency would be strengthened if the authors functionally tested this hypothesis. To that end, the in vitro stability assay would be useful, where the authors could address if increased mTORC1 signaling and glycolysis actually promotes Treg cell instability and increased cytokine production by Ndfip1-deficient Treg cells. These experiments could be performed using the published in vitro stability assays discussed above (similar to a recent study by Wei et al. 2016 Nature Immunology).

Minor comments:

6. In Figure 4g, it is misleading to label the graph as "Former Treg" when in the context of these mixed bone marrow chimeras, one cannot distinguish between Foxp3⁻ conventional T cells and Foxp3⁻ "ex"-Treg cells. Further, the authors should check that the labels in Supplemental Figure 1 are accurate and complete. The labels of "WT" are not specific enough to distinguish between the Foxp3-Cre⁺ males and the Foxp3-Cre^{+/-} females. Further, Figure S1A appears to be missing the label for the Foxp3-Cre^{+/-} x Ndfip1^{fl/fl} mice. Finally, the data in Figure 2f-l might be better representative of the conclusions if the authors presented each group as a separate column. Based on the authors' conclusions, it is presumed that the YFP-Foxp3⁺ cells from each mice are similar, but there are differences in the "deleted" YFP-Foxp3⁺ cell fractions between the two mice. The graphs, as shown, fail to definitely show a role independent of the inflammatory environment for these phenotypic changes.

7. The authors could also present Foxp3 mean fluorescence intensity values in their analysis of total Treg, cTreg, and eTreg cells. These data would help support the conclusions made later that Treg cell instability is affected by the loss of Ndfip1.

8. The authors state that Ndfip1 is important for maintaining Treg cell function that supports barrier function. However, the authors state that inflammation in other tissues aside from the skin, esophagus, and lung is minor. The intestines are often considered to be a barrier site; therefore, the authors should be more specific in their language when describing the data.

9. Survival data for their Foxp3-Cre⁺ male and Foxp3-Cre^{+/-} females would also be informative to the readers of this manuscript. It is presently unclear if the above mice develop disease with similar kinetics. The authors should also state in their methods at what ages the mice were analyzed and/or used to set up other experimental systems (e.g. mixed bone marrow chimeras, stability assays). Ideally, the cells used to set up secondary in vivo models would have been taken from mice that were not severely moribund.

10. Given their previous Nature Immunology study (Beal et al. 2012), it might be interesting to determine if Ndfip1 deficiency alerts peripherally-induced Treg cell homeostasis in sites like the lung and intestines of the Foxp3-Cre x Ndfip1^{fl/fl} mice.

REVIEWERS' COMMENTS:

Reviewer #1 (Remarks to the Author):

The authors have addressed my concerns. As a minor issue, the legend to Supp Fig. 8 states that cells were treated with '119 U/ml of rhIL-2 for 30minutes'. Is that the correct concentration?

Reviewer #2 (Remarks to the Author):

The authors have successfully addressed my questions.

Ndfip1 restricts TORC1 signaling and glycolysis in regulatory T cells to prevent autoinflammatory disease.

Response to reviewers' comments:

We thank the reviewers for their insightful review of this manuscript. Below are our point-by-point responses to their comments and concerns. In response to these comments, we have made significant improvements to the manuscript that clarify our findings and increase its overall impact. The key improvements include 1) our findings from 13 day old mice that the expansion of eTregs occurs very early, prior to measurable changes in IL-4 production by the cells and prior to the onset of inflammation (see new Suppl Fig 3), and 2) we clarify that the major consequence of the increased metabolism and proliferation in the cells is in allowing an accumulation of pathogenic IL-4 producing T cells. These additions were made possible by the reviewers comments and suggestions. In an effort to make our new data easy to find, added figures are highlighted below in yellow.

Reviewer #1 (Remarks to the Author):

Deng *et al.* study the functions of Ndfip1, an activator of Nedd4-family ubiquitin ligases, including Itch. While the Oliver lab had previously shown that Ndfip1 was important for the differentiation of Treg cells induced by TGF β , whether Ndfip1 is important for the functions or homeostasis of differentiated Treg cells had not been examined. Using a Foxp3-Cre deletion system, Deng *et al.* show that Treg-specific expression of Ndfip1 is important to avoid the expansion of the subset of Treg cells with effector functions (eTreg cells) and to promote stable Foxp3 expression. In addition, Ndfip1 represses IL-4 expression by Treg cells, as it does in conventional T cells, but IL-4 expression is not needed for its effect on eTreg amplification. Using metabolism (seahorse) analyses, the authors document increased glycolysis and glycolytic capacity in Ndfip1-deficient Treg cells, consistent with increased *in vitro* proliferation and acquisition of effector markers.

That manuscript is well presented and the data of high quality. It reports a comprehensive study of Ndfip1 functions in Treg cells, and provides insight into its impact on cell metabolism. Given the importance of these cells in immune responses, the data will be of interest to a large number of immunologists. A few suggestions below are intended to improve the manuscript.

Main comments

1-It is not clear whether Ndfip1 only serves to control the expansion of eTreg cells or if it also stabilizes cTreg cells or prevents their inappropriate conversion into eTreg cells. This could be addressed by *in vitro* analyses (as in Fig. 8) or adoptive transfer experiments (as in Fig. 5) performed from sorted cTreg cells.

*We agree that it is important to determine whether Ndfip1 uniquely affects eTreg cells or has an effect on both cTreg and eTreg cells. cTreg cells are thymically derived and undergo peripheral conversion into eTreg cells via stimulation through TCR or through ICOS¹. In our *in vitro* expanded Treg culture experiments, we fully activate all sorted total Treg (cTreg and eTreg) cells during the initial 48 hours of culture via their TCR (using α CD3 and α CD28). Thus, after 48 hours, all cells are CD44 high, activated or 'eTreg' cells. Results from experiments where we started with sorted YFP⁺ cTreg cells (n=3-5 per genotype, (see panel to left) were virtually identical to those in which we started with sorted YFP⁺ total Treg*

cells ($n=5-7$ per genotype). We therefore present all of that data pooled together in **Figure 8a**.

To address this comment in a broader manner, we reasoned that if *Ndfip1* were important for limiting the expansion of cTreg cells we would expect to see increased numbers of cTreg cells in *Ndfip1^{fl/fl} Foxp3-Cre⁺* animals, in contrast, if *Ndfip1* limits the conversion of cTreg cells into eTreg cells, we would expect the numbers of cTreg cells to be decreased. However, as shown in **Figure 2g**, we found that cTreg cells found in an *Ndfip1^{fl/fl} Foxp3-Cre⁺* animal are equal to the number found in a WT animal (*Ndfip1^{+/+} Foxp3-Cre⁺* animal). This indicates that in the absence of *Ndfip1*, we do not see loss of cTreg cells to suggest increased conversion of cTreg cells into eTreg cells.

To delve further, we also considered whether absolute numbers of cTreg cells were maintained at normal levels due to increased thymic output.

To test this, we examined thymic output in mixed chimera animals where WT (*Ndfip1^{+/+} Foxp3-Cre⁺*) and *Ndfip1^{fl/fl} Foxp3-Cre⁺* Treg cells develop in the same environment. We found no difference in total thymic output in a Rag KO animal reconstituted with a 1:1 mixture of both *Ndfip1*-sufficient and *Ndfip1*-deficient bone marrow (**Supplementary Fig. 1a**). Furthermore, when considering total thymic Treg cells, there was no evidence of an increase in Treg cells of *Ndfip1^{fl/fl} Foxp3-Cre⁺* bone marrow origin (**Supplementary Fig. 1b**). This indicates that the increase in eTreg, but not cTreg, cell number in the periphery is not driven by inappropriate conversion of cTreg cells to eTreg cells and there is no increased output of thymic cTreg cells to mask a peripheral loss of cTreg cells due to inappropriate conversion.

2-The impact of *Ndfip1* on metabolism could be direct or indirect, and the interpretation of the Seahorse data is a bit complicated by the experimental setup (as cells need to be amplified in culture). While there is no need to expand these metabolism analyses in the current report, the author's recent demonstration that *Ndfip* molecules promote the degradation of Jak kinases raises the obvious possibility that increased metabolism in *Ndfip1*-deficient cells results from increased cytokine signaling. Is there any evidence for increased Jak expression or Stat phosphorylation in the mutant Treg cells?

To test this we sorted Treg cells from *Ndfip1^{fl/fl} Foxp3-Cre⁺* or *Ndfip1^{+/+} Foxp3-Cre⁺* WT animals and rested them overnight without cytokine or TCR stimulation. We then re-stimulated these cells with IL-2 briefly to detect STAT5 phosphorylation. We found that both WT and *Ndfip1*-deficient Treg cells could efficiently respond to IL-2 by phosphorylating STAT5, however there was no difference between fold change of pSTAT5 levels or in absolute levels of pSTAT5 between WT and *Ndfip1*-deficient Treg cells (**Supplementary Fig. 8**). This is consistent with data previously published by our lab showing that CD4⁺ T cells from *Ndfip1^{fl/fl} CD4 Cre* animals do not show accumulation of JAK1 or pSTAT5, unlike CD4 T cells from *Ndfip1^{fl/fl} Ndfip2^{-/-} CD4 Cre⁺* cells (see **Figures 7b, d** in O'Leary, C. E. et al. (2016)).² Thus the loss of *Ndfip1* alone may be insufficient to drive an accumulation of JAK1, possibly due to compensation by *Ndfip2*.

Minor comments:

-Do the mutant mice die prematurely?

Our IACUC protocol does not define death as an endpoint. However, we have observed that by ~9 weeks of age the base of the ears in males and is marked by a reddening of the area. Over time, this dermatitis progresses, and the ears scab over and thicken. Eventually, animals then develop sores on their napes and sometimes on their backs. Beyond 16 weeks, the male mice are uniformly overtly sick with these skin sores. We are asked to treat or euthanize them by the Laboratory

Animal Facility when the dermatitis begins to ulcerate and scab over. Using this euthanasia timepoint as an endpoint, yes, the health and fitness of the $Ndfip1^{fl/fl}$ $Foxp3-Cre^+$ animals is severely compromised. The section describing the phenotype of the mice is now worded to reflect the progressiveness of the dermatitis.

-In figures legends or methods, it would be useful to state whether animals used in experiments are male or female. This can otherwise be confusing (e.g. Fig. 4, bone marrow chimeras where one assumes donors are male).

The donors for these experiments were male. We now note the genders in each figure.

-Lines 168 and following, the text does not really describe the data shown in figures 3h-l, and is confusing when compared to Figs. S2c-f. The text could be clarified to better fit with the actual figures.

We have now noted more clearly that Figure 3h-l describes the ratio of YFP+ to YFP- cells within a WT female animal compared to that same ratio within an $Ndfip1^{fl/fl}$ $Foxp3-Cre^{+/-}$ female animal. Therefore in each animal YFP+ cells are normalized to the YFP- cells from within the same animal (to normalize for an effect of Cre). This is in contrast to Figure S2c-f where we are comparing YFP- cells only from a WT, un-inflamed environment to YFP- cells from an inflamed $Ndfip1^{fl/fl}$ $Foxp3-Cre^{+/-}$ environment in order to determine the effects of an inflammatory environment on WT YFP- Treg cells.

-Line 261: the subtitle does not fit with the actual data (which shows that IL-4 is not necessary).

This has been changed from “IL-4 alone is insufficient to drive effector T_{reg} expansion in the absence of $Ndfip1$ ” to “IL-4 is dispensable for the expansion of $Ndfip1$ -deficient effector T_{reg} cells”.

-Line 294, *Lag3* is not the official gene symbol for Galectin3.

*This was an error in nomenclature. While *Lag3* is found on Treg cells and is important for Treg cell function, *Lgals3* is the official gene name for the β -galactoside binding protein, Galectin3, which was identified in our proteomics screens. We found only one report of the role of galectin3 in Treg cells³. Given that this represents a single study that identified Treg cells only based on CD4 and CD25 expression, reference to this protein has been removed.*

-In Fig. 7, what is the percentage of cells that remain $Foxp3^+$ at the end of the culture?

*In the experiments shown in Figure 7, there was no statistically relevant difference between the percentages of cells that remained $Foxp3^+$ from WT versus $Ndfip1$ -Treg cells after 7 days of expansion. For **Figure 7a-d**, final $Foxp3\%$ was $74.45\% \pm 8.03$ for $Ndfip1^{fl/fl}$ $Foxp3-Cre^+$ versus 74.8 ± 9.80 for $Ndfip1^{+/+}$ $Foxp3-Cre^+$, for an $n=4$ in 2 independent experiments. For **Figure 7e-h**, final $Foxp3\%$ was 88.1 ± 2.08 for $Ndfip1^{fl/fl}$ $Foxp3-Cre^+$ versus 86.43 ± 1.94 for $Ndfip1^{+/+}$ $Foxp3-Cre^+$ for an $n=6-8$ in 3 independent experiments. This information has now been included in the figure legend for figure 7.*

Reviewer #2 (Remarks to the Author):

In their manuscript, “*Ndfip1* limits mTORC1 activity in regulatory T cells to prevent loss of stability and autoinflammatory disease” Deng *et al.* investigated the role of the *Ndfip1*, a co-activator of Nedd4-family, in Treg cells by conditionally deleting *Ndfip1* within $Foxp3^+$ Treg cells (using $Foxp3-Cre$). The loss of *Ndfip1* expression within Treg cells disrupts immune homeostasis and leads to inflammation of the esophagus, skin, and lungs. This phenotype is linked to elevated IL-4 production by Treg cells, which appears to destabilize FoxP3 expression. Mechanistically, the authors demonstrated that mTORC1-related

metabolic reprogramming, especially glycolysis, accounts for Treg cell instability observed in the absence of *Ndfip1*. In general, this study nicely detailed how *Ndfip1* controls Treg cell homeostasis and function and provided new mechanistic understanding for how Treg cells can lose stability and acquire effector-like features that contribute to barrier surface pathology. Below, the authors may find comments on how to better support the overall conclusions of their manuscript:

1. As the authors state, many of the phenotypes described in the manuscript might be a secondary consequence of inflammation. To address this issue, the authors analyzed female mice that expressed a heterozygous *Foxp3-Cre* allele (denoted as *Foxp3-Cre*^{+/-} in the manuscript), but these mice appear to have inflammation similar to the male *Ndfip1* conditional knockout mice (Supplemental Figure S2). Moreover, the authors state that mixed bone marrow chimeras generated with *Foxp3-Cre* x *Ndfip1*^{fl/fl} bone marrow develop signs of dermatitis by 8 weeks post bone marrow transfer. Given these data, it is difficult to rule out a role for inflammation in the phenotypic changes of *Ndfip1*-deficient Treg cells. To better support their conclusion that the changes are mostly attributed to loss of *Ndfip1*, the authors could also analyze the female *Foxp3-Cre*^{+/-} mice prior to the onset of inflammation (e.g. 7-10 day old mice) for their cTreg/eTreg phenotypes, ICOS, GITR, PD-1, *Foxp3*, and CD25 expression, as well as IL-4 production. This analysis would also allow the authors to analyze if Treg cells are lost in the periphery before inflammation, which would be predicted based on their data linking *Ndfip1* to the maintenance of Treg cell stability.

In the mixed chimera setting and in the female animals, Ndfip1-deficient and Ndfip1-sufficient Treg cells developed in the same physiological environment and were exposed to the same inflammatory environment. In spite of this, only Ndfip1-deficient Treg cells showed the increase in eTreg cell frequencies, increase in proliferation of eTreg cells and increase in surface expression of GITR and ICOS on eTreg cells. This demonstrates that the loss of Ndfip1 in Treg cells leads to these changes and/or that the inflammatory environment may distinctly and uniquely influence Ndfip1-deficient Treg cells.

In the analyses of female animals (Figure 3g-l and Supplementary Fig 2c-f) we dissected the difference between Treg-intrinsic phenotypes and inflammation-dependent phenotypes. To clarify the results obtained from this strategy, we have now added a schematic (Supplementary Fig. 2g) and clarified wording of these sections.

*Furthermore, we examined lung cells from 13-day-old female mice, prior to the onset of inflammation. We looked at 13-old mice for technical feasibility of lung harvest and processing, and because this is prior to the onset of splenomegaly or dermatitis. We found that at 13 days there is no sign of overt inflammation in the mice: there is no increase in spleen size (Supplementary Fig. 3a), no increase in inflammation index (Supplementary Fig. 3b), no increase in total numbers of foxp3+ cells in the periphery (Supplementary Fig. 3c), and minimal cytokine production from CD4 T cells (Supplementary Fig. 3d) in the *Ndfip1*^{fl/fl} *Foxp3-Cre*^{+/-} females compared to WT females. *Ndfip1*-deficient Treg cells were more activated (eTreg cells) (Supplementary Fig. 3f) and displayed higher expression of ICOS and GITR (Supplementary Fig. 3g,h). This reinforces that these phenotypes are cell-intrinsic changes that occur prior to the onset of overt inflammation in *Ndfip1*^{fl/fl} *Foxp3-Cre*^{+/-} animals.*

2. It is quite interesting that the presence of wild-type Treg cells does not control the inflammation driven by Treg-specific deletion of *Ndfip1*. A similar phenomenon was also recently reported when *Pten* was conditionally deleted in Treg cells, where mixed bone marrow chimeras containing both wild-type and *Pten*-deficient Treg cells display evidence of inflammation (Shrestha *et al.* . 2015 Nat Immunol). The conclusions from that study were that heterozygous loss of *Pten* is sufficient to promote Treg cell

instability that ultimately led to autoimmunity. Is there any evidence the Foxp3-Cre x Ndfip1fl/+ show signs of inflammation or Treg cell dysfunction?

In our development of Ndfip1^{fl/fl} Foxp3-Cre⁺ animals, we have not observed signs of dermatitis in any Ndfip1^{fl/+} Foxp3-Cre⁺ animals. Unlike the tumor suppressor gene, PTEN, which has been shown to be haploinsufficient when deleted only in Treg cells^{4,5} and in several mouse models of disease such as prostate cancer,⁶ in neurocognitive studies^{7,8}, and even in non-primate animal models⁹, Ndfip1 appears to be haplosufficient. Supporting this, Ndfip1^{-/-} and Ndfip1^{fl/fl} CD4 Cre⁺ animals develop a more severe version of the disease we have identified in the Ndfip1^{fl/fl} Foxp3-Cre⁺ animals. However, Ndfip1^{+/-} [See Figure S2, Oliver et al., 2006¹⁰] and Ndfip1^{fl/+} CD4Cre⁺ animals do not develop disease and have a normal, healthy life span.

3. Because Ndfip1 deficiency might attenuate Treg cell survival or proliferation in the adoptive transfer model, the authors should validate that Treg cell stability is affected by an alternative method. For instance, their conditional knockout mice could be bred with a lineage tracing model (e.g. Rosa-linked LSL-reporter mice) and/or in vitro stability assays could be employed. These systems would allow the authors to better dissect out the roles of Treg cell proliferation, survival, and changes in Foxp3 expression to the apparent reductions of Treg cells after adoptive transfer.

To address the reviewer's concerns, we do not find that Ndfip1-deficient Treg cells fail to survive or proliferate in our adoptive transfer setting. We have now included data as (Supplementary Fig. 5f) showing that Ndfip1-deficient current and former Treg cells are actually more likely to expand in vivo. Therefore animals receiving Ndfip1-deficient Treg cells have more total CD45.2+ cells at the end of 13 weeks (Supplementary Fig. 5g). In vitro, our Treg expansion cultures support that we do not see decreased survival of the Ndfip1-deficient Treg cells. Instead, we observe great expansion and growth of these cells compared to their Ndfip1-sufficient counterparts as shown in Figure 8. Taken together, the increased fitness of these Ndfip1-deficient cells in vitro and in vivo provides support to the idea that the elevated mTORC1 signaling in these Ndfip1-deficient Treg cells may fuel better expansion in vivo.

Our barrier mouse facility does not allow import of strains from facilities other than Jax and Taconic, without re-derivation. We were thus limited in the Rosa reporters we could consider for these studies. We obtained Rosa reporter mice from Jackson [B6;129S6-Gt(ROSA)26Sortm14(CAG-tdTomato)Hze/J, Stock No: 007908] and bred them to our Ndfip1^{fl/fl} Foxp3-Cre⁺ animals. The Ndfip1^{fl/fl} Foxp3-Cre⁺ on this Rosa background do develop splenomegaly, dermatitis, and have activated CD4 Treg cell subsets. However, we did not find this Rosa reporter to be a faithful reporter of Foxp3, as we found upwards of 50% of CD4+ cells expressing RFP with or without simultaneous expression of YFP, even among our WT controls (Ndfip1^{+/+} Foxp3-Cre⁺). Of more concern, we found a significant number of CD4- T cells that expressed Rosa-tdtomato. Therefore, we could not interpret data from these mice.

4. Ndfip1-deficient Treg cells acquire the ability to produce effector cytokines, with IL-4 production being of particular interest in this study. The co-culture experiments in Figure 8h demonstrated that changes in mTORC1-related signaling appear to be independent of IL-4 signaling. However, the authors did not fully address if IL-4 could impact Treg cell instability independently of mTORC1 function. For instance, this laboratory clearly demonstrated a role for Ndfip1 in limiting IL-4 production that can suppress Foxp3 induction under iTreg cell conditions (Beal et al. . 2012 Nature Immunology). Therefore, it is possible that excessive IL-4 signaling by Treg cells could contribute to the instability of Ndfip1-deficient Treg cells. To test this, the authors could perform in vitro assays where they could determine if excess IL-4 exacerbates and blocking IL-4 reduces Treg cell instability in the presence and absence of Ndfip1. Such data would be useful

in further distinguishing their results from other studies, where elevated mTORC1 signaling is linked to excessive IL-17 and IFN-gamma production, but not IL-4 production (Park *et al.* . 2013 J Clin Invest).

IL-2 signaling drives STAT5 binding to the CNS2 region in the foxp3 locus in order to stabilize foxp3 gene expression in the presence of proinflammatory cytokines¹¹. Therefore, in vitro we tested the ability of Foxp3+ cells to maintain their Foxp3 identity (defined as percentage and MFI of Foxp3) in the presence of a stabilizing environment (IL-2) or in the presence of a destabilizing environment (IL-4 plus anti-IL-2). We found that Ndfip1-deficient Treg cells were equally stabilized by IL-2 and destabilized by IL-4 compared to their Ndfip1-sufficient Treg counterparts (Supplementary Fig. 4d,e). This suggested that the Ndfip1-deficient Treg cells were not uniquely sensitive to the destabilizing effects of IL-4. We propose that due to their increased metabolic fitness, Ndfip1-deficient Treg cells are better able to proliferate and to expand in vivo. The ability of these Treg cells to make IL-4 (due directly to the loss of Ndfip1) combined with their expansion (fueled by increased mTORC1) generates large amounts of IL-4 producing cells in vivo, leading to cell recruitment and tissue damage that exacerbates inflammatory cytokine production. Therefore Ndfip1-deficient Treg cells may be more likely to be exposed to conditions in vivo that result in their instability. It is interesting to consider whether there are in vivo tissue-specific expression patterns of IL-2 versus IL-4 that may lead to site-specific differences in the in vivo stability of the Ndfip1-deficient Treg cells. Statements to this effect have been added to the discussion.

5. The authors nicely demonstrate that mTORC1-related signaling and glycolysis is elevated in the absence of Ndfip1. The conclusion that mTORC1-mediated glycolysis destabilizes Treg cells in the context of Ndfip1 deficiency would be strengthened if the authors functionally tested this hypothesis. To that end, the in vitro stability assay would be useful, where the authors could address if increased mTORC1 signaling and glycolysis actually promotes Treg cell instability and increased cytokine production by Ndfip1-deficient Treg cells. These experiments could be performed using the published in vitro stability assays discussed above (similar to a recent study by Wei *et al.* . 2016 Nature Immunology).

*We carried out the 4-day in vitro assays using YFP+ sorted Treg cells from Ndfip1^{+/+}Foxp3-Cre⁺ and Ndfip1^{fl/fl}Foxp3-Cre⁺ animals, cultured on plate-bound α CD3/ α CD28 in RPMI medium containing β mercaptoethanol, penicillin-streptomycin, glutamate, non-essential amino acids, HEPES and sodium pyruvate, with low dose IL-2 (200U/ml) in the presence of 50nm rapamycin similar to the protocol used in Wei *et al.*, 's autophagy paper.*

In our hands, inhibition of mTORC1 with rapamycin did not increase Foxp3 stability (based on %FoxP3+ or Foxp3 MFI). At the end of our 4-day cultures, the majority of cells in culture were still Foxp3+ (~86% across all groups with no significant genotype differences). Additionally, blocking mTORC1 signaling did not change the IL-4 production by these cells. In these experiments, Ndfip1-deficient Treg cells showed low but detectable IL-4 production compared to WT Treg cells and this did not reach statistical significance.

Therefore increased mTORC1 may be insufficient to drive loss of Foxp3 at steady state unless there is concurrent proliferation and expansion of these IL-4 producing Ndfip1-deficient Treg cells. This may explain why under steady state conditions, we do not see a large enough population of Ndfip1-deficient Treg cells that have lost Foxp3 protein (Supplementary Fig. 6a-c) although we do see by bisulfite sequencing that these Tregs already show signs of instability (increased methylation) of the Foxp3 CNS2 locus (Figure 5a,b). Based on these data, we have now deemphasized the role on Torc1 in loss of FoxP3 throughout the manuscript and in the discussion, attribute this to autocrine IL-4.

This question was key to enabling us to think more deeply about the whether the mTORC1-driven metabolic fitness and the IL-4 production were two inter-related or independent processes. Thus, we thank the reviewer for these suggestions.

6. In Figure 4g, it is misleading to label the graph as “Former Treg” when in the context of these mixed bone marrow chimeras, one cannot distinguish between Foxp3- conventional T cells and Foxp3- “ex”-Treg cells.

Corrected. Thank you.

Further, the authors should check that the labels in Supplemental Figure 1 are accurate and complete. The labels of “WT” are not specific enough to distinguish between the Foxp3-Cre+ males and the Foxp3-Cre+/- females.

Corrected. Thank you.

Further, Figure S1A appears to be missing the label for the Foxp3-Cre+/- x Ndfip1fl/fl mice.

Corrected. Thank you.

Finally, the data in Figure 2f-1 (**Figure3 f-1?**) might be better representative of the conclusions if the authors presented each group as a separate column. Based on the authors’ conclusions, it is presumed that the YFP-Foxp3+ cells from each mice are similar, but there are differences in the “deleted” YFP+Foxp3+ cell fractions between the two mice. The graphs, as shown, fail to definitely show a role independent of the inflammatory environment for these phenotypic changes.

We have included a sample analysis where we show surface expression of GITR from YFP- and YFP+ eTreg cells from female Ndfip1^{fl/fl}Foxp3-Cre⁺ and Ndfip1^{+/+}Foxp3-Cre⁺ mice as four separate columns (see panel below). A dashed line joins cells from the same mouse. This is the same data shown in Figure 3j in two columns as YFP+ to YFP- ratios. A prominent negative effect of Cre on the cell surface expression on GITR is evident. In Ndfip1^{+/+}Foxp3-Cre⁺ WT animals, the presence of the YFP/Cre fusion protein results in decreased GITR MFI on the eTreg cells from these completely Ndfip1-sufficient controls animals.

In Ndfip1^{+/+}Foxp3-Cre⁺ WT animals, the presence of the YFP/Cre fusion protein results in decreased GITR MFI on the eTreg cells from these completely Ndfip1-sufficient controls animals.

In the Ndfip1^{fl/fl}Foxp3-Cre⁺ animals, we see that in spite of this negative Cre effect on YFP+ cells, Ndfip1-deficient YFP+ cells have equivalently high expression of GITR as YFP- cells from the same mice. Normalizing YFP+ to YFP- values for each parameter measured allows us to emphasize changes that are due to loss of Ndfip1 and not those due to Cre. We have now clarified our strategy for

normalizing for the effect of Cre in our analysis in the text and have included a schema in Supplementary Fig. 2g that emphasizes which of the surface markers that we found to be dependent on inflammation versus those that were intrinsic to Ndfip1 deficiency.

7. The authors could also present Foxp3 mean fluorescence intensity values in their analysis of total Treg, cTreg, and eTreg cells. These data would help support the conclusions made later that Treg cell instability is affected by the loss of Ndfip1.

This is now provided as (Supplementary Fig. 6) and discussed in the text of the manuscript.

8. The authors state that Ndfip1 is important for maintaining Treg cell function that supports barrier function. However, the authors state that inflammation in other tissues aside from the skin, esophagus, and lung is minor. The intestines are often considered to be a barrier site; therefore, the authors should be more specific in their language when describing the data.

The language has been fixed. We have removed vague references to ‘barrier’ sites.

9. Survival data for their Foxp3-Cre+ male and Foxp3-Cre+/- females would also be informative to the readers of this manuscript. It is presently unclear if the above mice develop disease with similar kinetics.

We have clarified in the text that for certain measured parameters eg cytokine production from all CD4T cells or for spleen weight, male and female animals show comparable levels of pathology. However dermatitis, which is the first overt sign of illness that we see in these mice, usually starts with the males first. At 9-16 weeks of age, males have much greater penetrance of the disease compared to females.

The authors should also state in their methods at what ages the mice were analyzed and/or used to set up other experimental systems (e.g. mixed bone marrow chimeras, stability assays). Ideally, the cells used to set up secondary in vivo models would have been taken from mice that were not severely moribund.

This has been fixed. Figure legends now include genders and ages of mice used.

10. Given their previous Nature Immunology study (Beal *et al.* . 2012), it might be interesting to determine if Ndfip1 deficiency affects peripherally-induced Treg cell homeostasis in sites like the lung and intestines of the Foxp3-Cre x Ndfip1fl/fl mice.

This is a really interesting question. Scientists have continued to debate whether any marker is sufficient for distinguishing between nTreg and iTreg cells¹².

To properly examine this question, our mice would need to be crossed onto an OTII Rag KO background so we can track iTreg generation in vivo. This would be something to explore in future experiments but is beyond the scope of this paper..

References

1. Smigielski, K. S. *et al.* CCR7 provides localized access to IL-2 and defines homeostatically distinct regulatory T cell subsets. *J. Exp. Med.* **211**, 121–36 (2014).
2. O’Leary, C. E. *et al.* Ndfip-mediated degradation of Jak1 tunes cytokine signalling to limit expansion of CD4+ effector T cells. *Nat. Commun.* **7**, 11226 (2016).
3. Ocklenburg, F. *et al.* UBD , a downstream element of FOXP3 , allows the identification of LGALS3 , a new marker of human regulatory T cells. 724–737 (2006). doi:10.1038/labinvest.3700432
4. Shrestha, S. *et al.* Treg cells require the phosphatase PTEN to restrain TH1 and TFH cell responses. *Nat. Immunol.* **16**, 178–187 (2015).
5. Huynh, A. *et al.* Control of PI(3) kinase in Treg cells maintains homeostasis and lineage stability. *Nat. Immunol.* **16**, 188–96 (2015).
6. Kwabi-Addo, B. *et al.* Haploinsufficiency of the Pten tumor suppressor gene promotes prostate cancer progression. *Proc. Natl. Acad. Sci. U. S. A.* **98**, 11563–8 (2001).
7. Napoli, E. *et al.* Mitochondrial dysfunction in Pten Haplo-insufficient mice with social deficits and repetitive behavior: Interplay between Pten and p53. *PLoS One* **7**, 1–19 (2012).
8. Page, D. T., Kutti, O. J., Prestia, C. & Sur, M. Haploinsufficiency for Pten and Serotonin transporter cooperatively influences brain size and social behavior. *Proc. Natl. Acad. Sci. U. S. A.* **106**, 1989–94 (2009).
9. Choorapokayil, S., Kuiper, R. V, de Bruin, A. & den Hertog, J. Haploinsufficiency of the genes encoding the tumor suppressor Pten predisposes zebrafish to hemangiosarcoma. *Dis. Model. Mech.* **5**, 241–247 (2012).
10. Oliver, P. M. *et al.* Ndfip1 protein promotes the function of itch ubiquitin ligase to prevent T cell activation and T helper 2 cell-mediated inflammation. *Immunity* **25**, 929–940 (2006).
11. Feng, Y. *et al.* Control of the inheritance of regulatory T cell identity by a cis element in the foxp3 locus. *Cell* **158**, 749–763 (2014).
12. Lin, X. *et al.* Advances in distinguishing natural from induced Foxp3+ regulatory T cells. *Int. J. Clin. Exp. Pathol.* **6**, 116–123 (2013).

Ndfip1 restricts TORC1 signalling and glycolysis in regulatory T cells to prevent autoinflammatory disease.

Response to reviewers' comments:

We thank the reviewers for their acceptance of this manuscript.

Reviewer #1 (Remarks to the Author):

The authors have addressed my concerns. As a minor issue, the legend to Supp Fig. 8 states that cells were treated with '119 U/ml of rhIL-2 for 30minutes'. Is that the correct concentration?

119U/ml rhIL-2 is what we actually used. It corresponds to 50ng/ml IL-2. We have added the ng/ml unit to the main text of the manuscript for readers who are more familiar with cytokine units in ng/ml.

Reviewer #2 (Remarks to the Author):

The authors have successfully addressed my questions.